# SAN: Inducing Metrizability of GAN with Discriminative Normalized Linear Layer

**Yuhta Takida**[1] **Masaaki Imaizumi**[2] **Takashi Shibuya**[1] **Chieh-Hsin Lai**[1]
**Toshimitsu Uesaka**[1] **Naoki Murata**[1] **Yuki Mitsufuji**[1,3]
[1]Sony AI, [2]The University of Tokyo, [3]Sony Group Corporation

## Abstract

Generative adversarial networks (GANs) learn a target probability distribution by optimizing a generator and a discriminator with minimax objectives. This paper addresses the question of whether such optimization actually provides the generator with gradients that make its distribution close to the target distribution. We derive *metrizable conditions*, sufficient conditions for the discriminator to serve as the distance between the distributions, by connecting the GAN formulation with the concept of sliced optimal transport. Furthermore, by leveraging these theoretical results, we propose a novel GAN training scheme called the Slicing Adversarial Network (SAN). With only simple modifications, a broad class of existing GANs can be converted to SANs. Experiments on synthetic and image datasets support our theoretical results and the effectiveness of SAN as compared to the usual GANs. We also apply SAN to StyleGAN-XL, which leads to a state-of-the-art FID score amongst GANs for class conditional generation on CIFAR10 and ImageNet 256×256. Our implementation is available on the project page https://ytakida.github.io/san/.

## 1 Introduction

Utilizing a generative adversarial network (GAN) (Goodfellow et al., 2014) is a popular approach for generative modeling, and GANs have achieved remarkable performance in image (Brock et al., 2019; Karras et al., 2019; 2021), audio (Kumar et al., 2019; Donahue et al., 2019; Kong et al., 2020), and video (Tulyakov et al., 2018; Hao et al., 2021) domains. The aim of GAN is to learn a target probability measure via a neural network, called a generator. To achieve this, a discriminator is introduced, and the generator and discriminator are optimized in a minimax way.

Here, we pose the question of whether GAN optimization actually makes the generator distribution close to the target distribution. For example, likelihood-based generative models such as variational autoencoders (Kingma & Welling, 2014; Higgins et al., 2017; Zhao et al., 2019; Takida et al., 2022), normalizing flows (Tabak & Vanden-Eijnden, 2010; Tabak & Turner, 2013; Rezende & Mohamed, 2015), and denoising diffusion probabilistic models (Ho et al., 2020; Song et al., 2020; Nichol & Dhariwal, 2021) are optimized via the principle of minimizing the exact Kullback–Leibler divergence or its upper bound (Jordan et al., 1999). For GANs, in contrast, solving the minimization problem with the optimal discriminator is equivalent to minimizing a specific dissimilarity (Goodfellow et al., 2014; Nowozin et al., 2016; Lim & Ye, 2017; Miyato et al., 2018). Furthermore, GANs have been analyzed on the basis of the optimal discriminator assumption (Chu et al., 2020). However, real-world GAN optimizations hardly achieve maximization (Fiez et al., 2022), and analysis of GAN optimization without that assumption remains a challenge.

To analyze and investigate GAN optimization without the optimality assumption, there are various approaches from the perspectives of training convergence (Mescheder et al., 2017; Nagarajan & Kolter, 2017; Mescheder et al., 2018; Sanjabi et al., 2018; Xu et al., 2020), loss landscape around the saddle points of minimax problems (Farnia & Ozdaglar, 2020; Berard et al., 2020), and the smoothness of optimization (Fiez et al., 2022). Although these studies have been insightful for GAN stabilization, there has been little discussion as to whether trained discriminators indeed provide generator optimization with gradients that reduce dissimilarities.

Table 1: Common GAN losses do not simultaneously satisfy all the sufficient conditions given in Theorem 5.3. Thus, we propose SAN to address *direction optimality*. Even if a direction $\omega$ is the maximizer of the inner problems $\mathcal{V}$, it does not satisfy *direction optimality* except in Wasserstein GAN (see Sec. 6). In Appx. H, it is empirically demonstrated that a discriminator trained on Wasserstein GAN tends not to satisfy *separability*. We put $*$ in the last column since *injectivity* is not directly affected by objective functions but by other factors (see Appx. H for detailed explanations).

|  | Direction optimality | Separability | Injectivity |
|---|:---:|:---:|:---:|
| Wasserstein GAN | ✓ | weak | $*$ |
| GAN (Hinge, Saturating, Non-saturating) | ✗ | ✓ | $*$ |
| SAN (Hinge, Saturating, Non-saturating) | ✓ | ✓ | $*$ |

In this paper, we provide a unique perspective on GAN optimization that helps to consider whether a discriminator is *metrizable*.

**Definition 1.1** (Metrizable discriminator). *Let $\mu_\theta$ and $\mu_0$ be measures. Given an objective function $\mathcal{J}(\theta; \cdot)$ for $\theta$, a discriminator $f$ is $(\mathcal{J}, \mathcal{D})$- or $\mathcal{J}$-metrizable for $\mu_\theta$ and $\mu_0$, if $\mathcal{J}(\theta; f)$ is minimized only with $\theta \in \arg\min_\theta \mathcal{D}(\mu_0, \mu_\theta)$ for a certain distance on measures, $\mathcal{D}(\cdot, \cdot)$.*

To evaluate the dissimilarity with a given GAN minimization problem $\mathcal{J}$, we are interested in other conditions besides the discriminator's optimality. Hence, we propose *metrizable conditions*, namely, *direction optimality*, *separability*, and *injectivity*, that induce a $\mathcal{J}$-metrizable discriminator. We first introduce a divergence, called functional mean divergence (FM$^*$) in Sec. 3 and connect the FM$^*$ with the minimization objective function of Wasserstein GAN. Then, we obtain the *metrizable conditions* for Wasserstein GAN by investigating Question 1.2. We provide an answer to this question in Sec. 4 by relating the FM$^*$ to the concept of sliced optimal transport (Bonneel et al., 2015; Kolouri et al., 2019). Then, in Sec. 5, we formalize the proposed conditions for Wasserstein GAN and further extend the result to generic GAN.

**Question 1.2.** *Under what conditions is FM$^*$ a distance?*

Based on the derived *metrizable conditions*, we propose the Slicing Adversarial Network (SAN) in Sec. 6. As seen in Table 1, we find that optimal discriminators for most existing GANs do not satisfy *direction optimality*. Hence, we develop a modification scheme for GAN maximization problems to enforce *direction optimality* on our discriminator. Owing to the scheme's simplicity, GANs can easily be converted to SANs. We conduct experiments to verify our perspective and demonstrate that SANs are superior to GANs in certain generation tasks on synthetic and image datasets. In particular, we confirmed that SAN improves the state-of-the-art FID for conditional generation with StyleGAN-XL (Sauer et al., 2022) on CIFAR10 and ImageNet 256×256 despite the simple modifications.

## 2 PRELIMINARIES

### 2.1 NOTATIONS

We consider a sample space $X \subseteq \mathbb{R}^{D_x}$ and a latent space $Z \subseteq \mathbb{R}^{D_z}$. Let $\mathcal{P}(X)$ be the set of all probability measures on $X$, and let $L^\infty(X, \mathbb{R}^D)$ denote the $L^\infty$ space for functions $X \to \mathbb{R}^D$. Let $\mu$ (or $\nu$) represent a probability measure with probability density function $I_\mu$ (or $I_\nu$). Denote $d_h(\mu, \nu) := \mathbb{E}_{x \sim \mu}[h(x)] - \mathbb{E}_{x \sim \nu}[h(x)]$. We use the notation of the pushforward operator $\sharp$, which is defined as $g_\sharp \sigma := \sigma(g^{-1}(B))$ for $B \in X$ with a function $g : Z \to X$ and a probability measure $\sigma \in \mathcal{P}(Z)$. We denote the Euclidean inner product by $\langle \cdot, \cdot \rangle$. Lastly, $\hat{(\cdot)}$ denotes a normalized vector. Please also refer to Appx. A for some notations defined in the rest of the papers.

### 2.2 PROBLEM FORMULATION IN GANs

Assume that we have data obtained by discrete sampling from a target probability distribution $\mu_0 \in \mathcal{P}(X)$. Then, we introduce a trainable generator function $g_\theta : Z \to X$ with parameter $\theta \in \mathbb{R}^{D_\theta}$ to model a trainable probability measure as $\mu_\theta = g_{\theta\sharp}\sigma$ with $\sigma \in \mathcal{P}(Z)$. The aim of generative modeling here is to learn $g_\theta$ so that it approximates the target measure as $\mu_\theta \approx \mu_0$.

For generative modeling in GAN, we introduce the notion of a discriminator $f \in \mathcal{F}(X) \subset L^{\infty}(X, \mathbb{R})$. We formulate the GAN's optimization problem as a two-player game between the generator and discriminator with $\mathcal{V} : \mathcal{F}(X) \times \mathcal{P}(X) \to \mathbb{R}$ and $\mathcal{J} : \mathbb{R}^{D_\theta} \times \mathcal{F}(X) \to \mathbb{R}$, as follows:

$$\max_{f \in \mathcal{F}(X)} \mathcal{V}(f; \mu_\theta) \quad \text{and} \quad \min_{\theta \in \mathbb{R}^{D_\theta}} \mathcal{J}(\theta; f). \tag{1}$$

Regarding the choices of $\mathcal{V}$ and $\mathcal{J}$, there are GAN variants (Goodfellow et al., 2014; Nowozin et al., 2016; Arjovsky et al., 2017) that lead to different dissimilarities between $\mu_0$ and $\mu_\theta$ with the maximizer $f$. In this paper, we use a representation of the discriminator in an inner-product form:

$$f(x) = \langle \omega, h(x) \rangle, \tag{2}$$

where $\omega \in \mathbb{S}^{D-1}$ and $h(x) \in L(X, \mathbb{R}^D)$. The form is naturally represented by a neural network[1].

## 2.3 WASSERSTEIN DISTANCE AND ITS USE FOR GANS

We consider the Wasserstein-$p$ distance (Villani, 2009) between probability measures $\mu$ and $\nu$

$$W_p(\mu, \nu) := \left( \inf_{\pi \in \Pi(\mu, \nu)} \int_{X \times X} \|x - x'\|_p^p d\pi(x, x') \right)^{\frac{1}{p}}, \tag{3}$$

where $p \in [1, \infty)$ and $\Pi(\mu, \nu)$ is the set of all coupling measures whose marginal distributions are $\mu$ and $\nu$. The idea of Wasserstein GAN is to learn a generator by minimizing the Wasserstein-1 distance between $\mu_0$ and $\mu_\theta$. For this goal, one could adopt the Kantorovich—Rubinstein (KR) duality representation to rewrite Eq. (3) and obtain the following optimization problem:

$$\max_{f \in \mathcal{F}_{\text{Lip}}(X)} \mathcal{V}_{\text{W}}(f; \mu_\theta) := d_f(\mu_0, \mu_\theta) \tag{4}$$

where $\mathcal{F}_{\text{Lip}}$ denotes the class of 1-Lipschitz functions. In contrast, we formulate an optimization problem for the generator as a minimization of the right side of Eq. (4) w.r.t. the generator parameter:

$$\min_{\theta \in \mathbb{R}^{D_\theta}} \mathcal{J}_{\text{W}}(\theta; f) := -\mathbb{E}_{x \sim \mu_\theta}[f(x)]. \tag{5}$$

## 2.4 SLICED OPTIMAL TRANSPORT

The Wasserstein distance is highly intractable when the dimension $D_x$ is large (Arjovsky et al., 2017). However, it is well known that *sliced optimal transport* can be applied to break this intractability by projecting the data on a one-dimensional space. That is, for the $D_x = 1$ case, the Wasserstein distance has a closed-form solution:

$$W_p(\mu, \nu) = \left( \int_0^1 |F_\mu^{-1}(\rho) - F_\nu^{-1}(\rho)|^p d\rho \right), \tag{6}$$

where $F_\mu^{-1}(\cdot)$ denotes the quantile function for $I_\mu$. The closed-form solution for a one-dimensional space prompted the emergence of the concept of sliced optimal transport.

In the original sliced Wasserstein distance (SW) (Bonneel et al., 2015), a probability density function $I$ on the data space $X$ is mapped to a probability density function of $\xi \in \mathbb{R}$ by the standard Radon transform (Natterer, 2001; Helgason, 2011) as $\mathcal{R}I(\xi, \omega) := \int_X I(x)\delta(\xi - \langle x, \omega \rangle)dx$, where $\delta(\cdot)$ is the Dirac delta function and $\omega \in \mathbb{S}^{D_x-1}$ is a direction. The sliced Wasserstein distance between $\mu$ and $\nu$ is defined as $SW_p^{h,\omega}(\mu, \nu) := (\int_{\omega \in \mathbb{S}^{D_x-1}} W_p^p(\mathcal{R}I_\mu(\cdot, \omega), \mathcal{R}I_\nu(\cdot, \omega))d\omega)^{1/P}$. Intuitively, the idea behind this distance is to decompose high-dimensional distributions into an infinite number of pairs of tractable distributions by linear projections.

Various extensions of the sliced Wasserstein distance have been proposed (Kolouri et al., 2019; Deshpande et al., 2019; Nguyen et al., 2021). Here, we review an extension called augmented sliced Wasserstein distance (ASW) (Chen et al., 2022). Given a measurable injective function $h : X \to$

---

[1]In practice, the discriminator $f$ is implemented as in Eq. (2), e.g., $f_\phi(x) = w_{\phi_L}^\top(l_{\phi_{L-1}} \circ l_{\phi_{L-2}} \circ \cdots \circ l_{\phi_1})(x)$ with nonlinear layers $\{l_{\phi_\ell}\}_{\ell=1}^{L-1}$, $w_{\phi_L} \in \mathbb{R}^D$, and their weights $\phi := \{\phi_\ell\}_{\ell=1}^{L} \in \mathbb{R}^{d_\phi}$.

$\mathbb{R}^D$, the distance is obtained via the spatial Radon transform (SRT), which is defined for any $\xi \in \mathbb{R}$ and $\omega \in \mathbb{S}^{d-1}$, as follows:

$$\mathcal{S}^h I(\xi, \omega) := \int_X I(x)\delta(\xi - \langle \omega, h(x) \rangle)dx. \tag{7}$$

The ASW-$p$ is then obtained via the SRT in the same fashion as the standard SW:

$$ASW_p^h(\mu, \nu) := \left( \int_{\omega \in \mathbb{S}^{d-1}} W_p^p(\mathcal{S}^h I_\mu(\cdot, \omega), \mathcal{S}^h I_\nu(\cdot, \omega))d\omega \right)^{\frac{1}{p}}. \tag{8}$$

Equation (6) can be used to evaluate the integrand in Eq. (8), which is usually evaluated via approximated quantile functions with sorted finite samples from $\mathcal{S}^h I_\nu(\cdot, \omega)$ and $\mathcal{S}^h I_\nu(\cdot, \omega)$.

## 3 FORMULATION OF QUESTION 1.2

We introduce a divergence called the functional mean divergence, FM or FM$^*$, which is defined for a given functional space or function, respectively. Minimization of the FM$^*$ can be formulated as an optimization problem involving $\mathcal{J}_W$, and we cast Question 1.2 in this context. In Sec. 4, we provide an answer to this question, which in turn provides the *metrizable conditions* with $\mathcal{J}_W$ in Sec. 5.

### 3.1 PROPOSED FRAMEWORK: FUNCTIONAL MEAN DIVERGENCE

We start by defining the FM with a given functional space.

**Definition 3.1** (Functional Mean Divergence (FM)). *We define a family of FMs as*

$$\mathscr{D}_D^{FM} := \left\{ (\mu, \nu) \mapsto \max_{h \in \mathcal{F}(X)} \|d_h(\mu, \nu)\|_2 \,|\, \mathcal{F}(X) \subseteq L^\infty(X, \mathbb{R}^D) \right\}, \tag{9}$$

*Further, we denote an instance in the family as $FM_\mathcal{F}(\mu, \nu) \in \mathscr{D}_D^{FM}$, where $\mathcal{F}(X) \subseteq L^\infty(X, \mathbb{R}^D)$.*

By definition, the FM family includes the integral probability metric (IPM) (Müller, 1997), which includes the Wasserstein distance in KR form as a special case.

**Proposition 3.2.** *For $\mathcal{F}(X) \in L^\infty(X, \mathbb{R})$, $IPM_\mathcal{F}(\cdot, \cdot) := \max_{f \in \mathcal{F}} d_f(\cdot, \cdot) \in \mathscr{D}_1^{FM}$.*

The FM is an extension of the IPM to deal with vector-valued functional spaces. Although the FM with a properly selected functional space yields a distance between target distributions, the maximization in Eq. (9) is generally difficult to achieve. Instead, we use the following metric, which is defined for a given function.

**Definition 3.3** (Functional Mean Divergence$^*$ (FM$^*$)). *Given a functional space $\mathcal{F}(X) \subseteq L^\infty(X, \mathbb{R}^D)$, we define a family of FMs$^*$ as*

$$\mathscr{D}_\mathcal{F}^{FM^*} := \{ (\mu, \nu) \mapsto \|d_h(\mu, \nu)\|_2 \,|\, h \in \mathcal{F}(X) \}. \tag{10}$$

*Further, we denote an instance in the family as $FM_h^*(\mu, \nu) \in \mathscr{D}_\mathcal{F}^{FM^*}$, where $h \in \mathcal{F}(X)$.*

We are interested in Question 3.4, which is a mathematical formulation of Question 1.2. We give an answer to this question in Sec. 4. Since optimization of the FM$^*$ is related to $\mathcal{J}_W$ in Sec. 3.2, the conditions in Question 3.4 enable us to derive $(\mathcal{J}_W, FM_h^*)$-*metrizable conditions* in Sec. 5.

**Question 3.4.** *Under what conditions for $\mathcal{F}(X) \in L^\infty(X, \mathbb{R}^D)$ are $FM_h^*(\cdot, \cdot) \in \mathscr{D}_\mathcal{F}^{FM^*}$ distances?*

### 3.2 DIRECTION OPTIMALITY TO CONNECT FM$^*$ AND $\mathcal{J}_W$

Optimization of the FM$^*$ with a given $h \in L^\infty(X, \mathbb{R}^D)$ returns us to an optimization problem involving $\mathcal{J}_W$.

**Proposition 3.5** (*Direction optimality* connects FM$^*$ and $\mathcal{J}_W$). *Let $\omega$ be on $\mathbb{S}^{D-1}$. For any $h \in L^\infty(X, \mathbb{R}^D)$, minimization of $FM_h^*(\mu_\theta, \mu_0)$ is equivalent to optimization of $\min_{\theta \in \mathbb{R}^{D_\theta}} \max_{\omega \in \mathbb{S}^{D-1}} \mathcal{J}_W(\theta; \langle \omega, h \rangle)$. Thus, $\nabla_\theta FM_h^*(\mu_\theta, \mu_0) = \nabla_\theta \mathcal{J}_W(\theta; \langle \omega^*, h \rangle)$, where $\omega^*$ is the optimal solution (direction) given as follows:*

$$\omega^* = \arg\max_{\omega \in \mathbb{S}^{D-1}} d_{\langle \omega, h \rangle}(\mu_0, \mu_\theta). \tag{11}$$

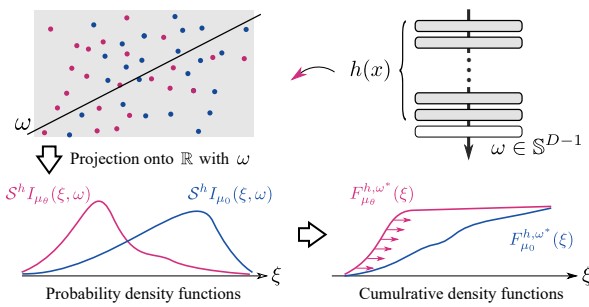
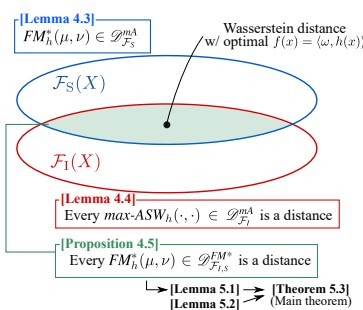

Figure 1: Discriminator decomposition into inner-product form $\langle \omega, h(x) \rangle$. Direction $\omega$ projects $h(x)$ onto $\mathbb{R}$. In this figure, $h$ is *separable* because $F_{\mu_0}^{h,\omega^*}(\xi) \leq F_{\mu_\theta}^{h,\omega^*}(\xi)$ for all $\xi \in \mathbb{R}$ (see Definition 4.2).

Figure 2: Outline of Sec. 4. Proposition 4.5 is a major step toward our main theorem.

The proof can be found in Appx. D, where the key idea involves using the Cauchy-Schwartz inequality, $\|d_h(\mu_0, \mu_\theta)\|_2 \geq \langle \omega, d_h(\mu_0, \mu_\theta) \rangle$ with $\omega \in \mathbb{S}^{D-1}$, and the linearity of $d_h(\mu_0, \mu_\theta)$, leading to $\|d_h(\mu_0, \mu_\theta)\|_2 \geq d_{\langle \omega, h \rangle}(\mu_0, \mu_\theta)$.

Recall that we formulated the discriminator in the inner-product form (2), which is aligned with Proposition 3.5. Here, we introduced a notion of *optimal direction*, which most distinguishes $\mu_0$ and $\mu_\theta$ in the feature space given by $h$, intuitively (refer to Figure 1 for illustration). We refer to the condition for the direction in Eq. (11) as *direction optimality*. It is obvious here that, given function $h$, the maximizer $\omega^*$ becomes $\hat{d}_h(\mu_0, \mu_\theta)$.

From the discussion in this section, Proposition 3.5 supports the notion that investigating Question 3.4 will reveal the $(\mathcal{J}_W, FM_h^*)$-*metrizable conditions*.

## 4 CONDITIONS FOR METRIZABILITY: ANALYSIS BY MAX-ASW

### 4.1 STRATEGY FOR ANSWERING QUESTION 3.4

We consider the conditions of $\mathcal{F}(X)$ for Question 3.4 in the context of sliced optimal transport. To this end, we define a variant of sliced optimal transport, called maximum augmented sliced Wasserstein divergence (max-ASW) in Definition 4.1. In Sec. 4.2, we first introduce a condition called the *separable* condition, where divergences included in the FM family are also included in the max-ASW family. In Sec. 4.3, we further introduce a condition called the *injective* condition, where the max-ASW is a distance. Finally, imposing these conditions on $\mathcal{F}(X)$ brings us the desired conditions (see Fig. 2 for the discussion flow).

**Definition 4.1** (Maximum Augmented Sliced Wasserstein Divergence (max-ASW)). *Given a functional space $\mathcal{F}(X) \subseteq L^\infty(X, \mathbb{R}^D)$, we define a family of max-ASWs as*

$$\mathscr{D}_{\mathcal{F}}^{mA} := \left\{ (\mu, \nu) \mapsto \max_{\omega \in \mathbb{S}^{D-1}} W_1 \left( \mathcal{S}^h I_\mu(\cdot, \omega), \mathcal{S}^h I_\nu(\cdot, \omega) \right) | h \in \mathcal{F}(X) \right\}. \tag{12}$$

*Further, we denote an instance of the family as max-ASW$_h(\mu, \nu) \in \mathscr{D}_{\mathcal{F}}^{mA}$, where $h \in \mathcal{F}(X)$.*

Note that although the formulation in the definition includes the SRT, similarly to the ASW in Eq. (8), they differ in terms of the method of direction sampling ($\omega$).

### 4.2 *Separability* FOR EQUIVALENCE OF FM* AND MAX-ASW

We introduce a property for the function $h$, called *separability*, which connects the FM* and the max-ASW as in Lemma 4.3.

**Definition 4.2** (Separable). *Given $\mu, \nu \in \mathcal{P}(X)$, let $\omega$ be on $\mathbb{S}^{D-1}$, and let $F_\mu^{h,\omega}(\cdot)$ be the cumulative distribution function of $\mathcal{S}^h I_\mu(\cdot, \omega)$. If $\omega^* = \hat{d}_h(\mu, \nu)$ satisfies $F_\mu^{h,\omega^*}(\xi) \leq F_\nu^{h,\omega^*}(\xi)$ for any $\xi \in \mathbb{R}$, $h \in L^\infty(X, \mathbb{R}^D)$ is separable for those probability measures. We denote the class of all these separable functions for them as $\mathcal{F}_{S(\mu,\nu)}(X)$ or $\mathcal{F}_S$ for notation simplicity.*

$$\underset{\text{Wasserstein GAN loss}}{\min_\theta \mathcal{J}_W(\theta, \langle \omega^*, h \rangle)} \approx \min_\theta FM_h^*(\mu_\theta, \mu_0) \approx \underset{\substack{\textit{Injectivity} \text{ on } h \text{ ensures}\\ \textit{max-ASW} \text{ is a distance}}}{\min_\theta max\text{-}ASW_h(\mu_\theta, \mu_0)}$$

*Direction optimality* on $\omega$     *Separability* on $h$

Figure 3: An example with Wasserstein GAN loss: the metrizable conditions (*direction optimality*, *separability*, and *injectivity*) ensure that Wasserstein GAN loss evaluates the distance between data and generator distributions.

**Lemma 4.3.** *Given $\mu, \nu \in \mathcal{P}(X)$, every $h \in \mathcal{F}_{S(\mu,\nu)}(X)$ satisfies $FM_h^*(\mu, \nu) \in \mathscr{D}_{\mathcal{F}_S}^{mA}$.*

Intuitively, *separability* ensures that optimal transport maps for all the samples from $\mathcal{S}^h I_{\mu_\theta}(\cdot, \omega)$ to $\mathcal{S}^h I_{\mu_0}(\cdot, \omega)$ share the same direction as in Figure 1. For a general function $h \in L^\infty(X, \mathbb{R}^D)$, $FM_h^*(\cdot, \cdot)$ is not necessarily included in the max-ASW family, i.e., $\mathcal{D}(\cdot, \cdot) \in \mathscr{D}_{\mathcal{F}}^{mA}(\cdot, \cdot) \not\Leftrightarrow \mathcal{D}(\cdot, \cdot) \in \mathscr{D}_{\mathcal{F}}^{FM^*}$ for general $\mathcal{F}(X) \subseteq L^\infty(X, \mathbb{R}^D)$. Given $h$, calculation of the max-ASW via Eq. (6) generally involves evaluating the sign of the difference between the quantile functions. Intuitively, the equivalence between the FM and max-ASW distances holds if the sign is always positive regardless of $\rho \in [0, 1]$; otherwise, the sign's dependence on $\rho$ breaks the equivalence.

### 4.3 INJECTIVITY FOR MAX-ASW TO BE A DISTANCE

Imposing *injectivity* on $h$ guarantees that the induced max-ASW is indeed a distance. Intuitively, *injectivity* prevents the loss of information from the original samples $x \sim \mu_\theta$ and $x \sim \mu_0$.

**Lemma 4.4.** *Every max-ASW$_h(\cdot, \cdot) \in \mathscr{D}_{\mathcal{F}_I}^{mA}$ is a distance, where $\mathcal{F}_I$ indicates a class of all the injective functions in $L^\infty(X, \mathbb{R}^D)$.*

Lemmas 4.3 and 4.4immediately tell us that $FM_h^*(\mu, \nu)$ with a *separable* and *injective* $h$ is indeed a distance because it is included in the family of max-ASW *distances*, as stated in Proposition 4.5. With the result, we now have one of the answers to Question 3.4, which hints at our main theorem.

**Proposition 4.5.** *Given $\mu, \nu \in \mathcal{P}(X)$, every $FM_h^*(\mu, \nu) \in \mathscr{D}_{\mathcal{F}_{I,S}}^{FM^*}$ is indeed a distance, where $\mathcal{F}_{I,S}$ indicates $\mathcal{F}_I \cap \mathcal{F}_S$.*

## 5 DOES GAN TRAINING ACTUALLY MINIMIZE DISTANCE?

We present Theorem 5.3, which is our main theoretical result and gives sufficient conditions for the discriminator to be $\mathcal{J}$-metrizable. This theorem inspires us to propose modification schemes for GAN training in Sec. 6.

We directly apply the discussion in the previous section to $\mathcal{J}_W$, and by extending the result for Wasserstein GAN to a general GAN, we derive the main result. First, a simple combination of Propositions 3.5 and 4.5 yields the following lemma. Lemma 5.1 provides the conditions for the discriminator to be $\mathcal{J}_W$-*metrizable* (see Fig. 3 for how each condition works).

**Lemma 5.1** ($\mathcal{J}_W$-metrizable). *Given $h \in \mathcal{F}_I \cap \mathcal{F}_S$ with $\mathcal{F}_I, \mathcal{F}_S \subseteq L^\infty(X, \mathbb{R}^D)$, let $\omega^* \in \mathbb{S}^{D-1}$ be $\hat{d}_h(\mu_0, \mu_\theta)$. Then $f(x) = \langle \omega, h(x) \rangle$ is $(\mathcal{J}_W, FM_h^*)$-metrizable.*

Next, we generalize this result to more generic minimization problems. Our scope here is the minimization problems of general GANs that are formalized in the form $\mathcal{J}(\theta; f) = \mathbb{E}_{x \sim \mu_\theta}[R_\mathcal{J} \circ f(x)]$ with $R_\mathcal{J} : \mathbb{R} \to \mathbb{R}$. We utilize the gradient of such minimization problems w.r.t. $\theta$:

$$\nabla_\theta \mathcal{J}(\theta; f) = -\mathbb{E}_{z \sim \sigma}\left[r_\mathcal{J} \circ f(g_\theta(z)) \nabla_\theta f(g_\theta(z))\right], \tag{13}$$

where $r_\mathcal{J}(\cdot)$ is the derivative of $R_\mathcal{J}(\cdot)$, as listed in Table 2. By ignoring a scaling factor, Eq. (13) can be regarded as a gradient of $d_f(\tilde{\mu}_0^{r_\mathcal{J} \circ f}, \tilde{\mu}_\theta^{r_\mathcal{J} \circ f})$, where $\tilde{\mu}^{r \circ f}$ is defined via $I_{\tilde{\mu}^{r \circ f}}(x) \propto r \circ f(x) I_\mu(x)$. To examine whether updating the generator with the gradient in Eq. (13) can minimize a certain distance between $\mu_0$ and $\mu_\theta$, we introduce the following lemma.

**Lemma 5.2.** *For any $\rho : X \to \mathbb{R}_+$ and a distance for probability measures $\mathcal{D}(\cdot, \cdot)$, $\mathcal{D}(\tilde{\mu}^\rho, \tilde{\nu}^\rho)$ indicates a distance between $\mu$ and $\nu$.*

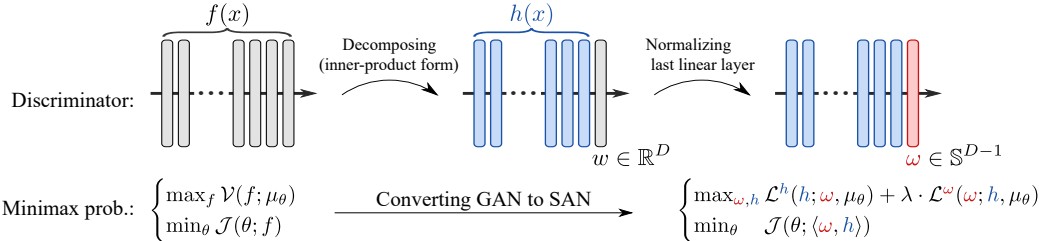

Figure 4: Converting GAN to SAN requires only simple modifications to discriminators.

We finally derive our main result, the $\mathcal{J}$-*metrizable conditions* for $\mu_0$ and $\mu_\theta$, by leveraging Lemma 5.2 and applying Propositions 3.5 and 4.5 to $\tilde{\mu}_0^{r_{\mathcal{J}} \circ f}$ and $\tilde{\mu}_\theta^{r_{\mathcal{J}} \circ f}$, The theorem suggests that the discriminator can serve as a distance between the generator and target distributions even if it is not the optimal solution to the original maximization problem $\mathcal{V}$.

**Theorem 5.3** ($\mathcal{J}$-Metrizability). *Given a functional $\mathcal{J}(\theta; f) := \mathbb{E}_{x \sim \mu_\theta}[R \circ f(x)]$ with $R : \mathbb{R} \to \mathbb{R}$ whose derivative, denoted as $R'$, is positive, let $h \in L^\infty(X, \mathbb{R}^D)$ and $\omega \in \mathbb{S}^{D-1}$ satisfy the following conditions. Then, $f(x) = \langle \omega, h(x) \rangle$ is $\mathcal{J}$-metrizable for $\mu_\theta$ and $\mu_0$.*

- *(**Direction optimality**) $\omega = \arg \max_{\tilde{\omega} \in \mathbb{S}^{D-1}} d_{\langle \tilde{\omega}, h \rangle}(\tilde{\mu}_0^{R' \circ f}, \tilde{\mu}_\theta^{R' \circ f})$,*
- *(**Separability**) $h$ is separable for $\tilde{\mu}_0^{R' \circ f}$ and $\tilde{\mu}_\theta^{R' \circ f}$,*
- *(**Injectivity**) $h$ is an injective function.*

In particular, *direction optimality* is the most interesting aspect for investigating GAN among the three conditions. It should be noted that most existing GANs besides Wasserstein GAN do not satisfy *direction optimality* with the maximizer $\omega$ of $\mathcal{V}$, which will be explained in Sec. 6. Therefore, we will focus on *direction optimality* in the next section. Please refer to Appx. H for a more detailed explanation and empirical investigation about the other conditions. Our observations regarding inductive effects of GAN losses to the metrizable conditions are summarized in Table 1. Note that *injectivity* cannot be directly controlled by loss designs as mentioned in the table.

## 6 SLICING ADVERSARIAL NETWORK

This section describes our proposed model, the Slicing Adversarial Network (SAN) to achieve the *direction optimality* of $\omega$ in Theorem 5.3 while keeping *separability* of $h$. We develop SAN by modifying the maximization problem $\mathcal{V}$ to guarantee that the optimal solution $\omega$ achieves *direction optimality*. The proposed modification scheme is applicable to most existing GAN objectives and does not necessitate additional exhaustive hyperparameter tuning or computational complexity.

As mentioned in Sec. 5, given a function $h$, the maximization problems in most GANs (excluding Wasserstein GAN) cannot achieve *direction optimality* with the maximum solution of $\mathcal{V}$ (see Table 1). We use hinge GAN as an example to illustrate this claim. The objective function to be maximized in hinge GAN is formulated as

$$\mathcal{V}_{\text{Hinge}}(\langle \omega, h \rangle; \mu_\theta) := \mathbb{E}_{x \sim \mu_0}[\min(0, -1 + \langle \omega, h(x) \rangle)] + \mathbb{E}_{x \sim \mu_\theta}[\min(0, -1 - \langle \omega, h(x) \rangle)]. \quad (14)$$

Given $h$, the maximizer $\omega$ becomes $\hat{d}_h(\mu_0^{\text{tr}}, \mu_\theta^{\text{tr}})$, where $\mu_0^{\text{tr}}$ and $\mu_\theta^{\text{tr}}$ denote truncated distributions whose supports are restricted by conditioning $x$ on $\langle \omega, h(x) \rangle < 1$ and $\langle \omega, h(x) \rangle > -1$, respectively. Since $(\mu_0^{\text{tr}}, \mu_\theta^{\text{tr}})$ is generally different from $(\tilde{\mu}_0^{r_{\mathcal{J}}}, \tilde{\mu}_\theta^{r_{\mathcal{J}}}) = (\mu_0, \mu_\theta)$, the maximum solution does not satisfy *direction optimality* for $\tilde{\mu}_0^{r_{\mathcal{J}}}$ and $\tilde{\mu}_\theta^{r_{\mathcal{J}}}$.

In line with the above discussion, we propose the following novel maximization problem:

$$\max_{\omega \in \mathbb{S}^{d-1}, h \in \mathcal{F}(X)} \mathcal{V}^{\text{SAN}}(\omega, h; \mu_\theta) := \underbrace{\mathcal{V}(\langle \omega^-, h \rangle; \mu_\theta)}_{\mathcal{L}^h(h; \omega, \mu_\theta)} + \lambda \cdot \underbrace{d_{\langle \omega, h^- \rangle}(\tilde{\mu}_0^{r_{\mathcal{J}} \circ f}, \tilde{\mu}_\theta^{r_{\mathcal{J}} \circ f})}_{\mathcal{L}^\omega(\omega; h, \mu_\theta)}, \quad (15)$$

where $(\cdot)^-$ indicates a stop-gradient operator. The first and second terms induce *separability* on $h$ and *direction optimality* on $\omega$, respectively. We simply set $\lambda \in \mathbb{R}_+$ to 1 in our experiments. The proposed modification scheme in Eq. (15) enables us to select any maximization objective for $h$ with

Table 2: Minimization problem and weighting function for direction optimization.

| | Minimization problem $\mathcal{J}$ | Weighting $r_{\mathcal{J}} \circ f(x)$ |
|---|---|---|
| Wasserstein GAN / Hinge GAN | $-\mathbb{E}_{x \sim \mu_\theta}\left[f(x)\right]$ | 1 |
| Saturating GAN | $-\mathbb{E}_{x \sim \mu_\theta}\left[\log \varsigma(f(x))\right]$ | $1 - \varsigma(f(x))$ |
| Non-saturating GAN | $\mathbb{E}_{x \sim \mu_\theta}\left[\log \varsigma(1 - f(x))\right]$ | $\varsigma(f(x))$ |

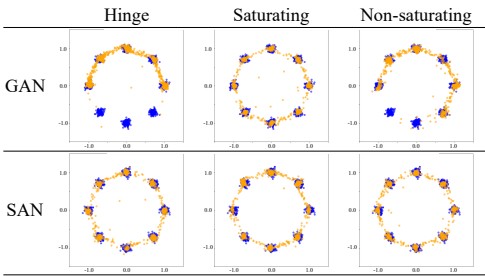

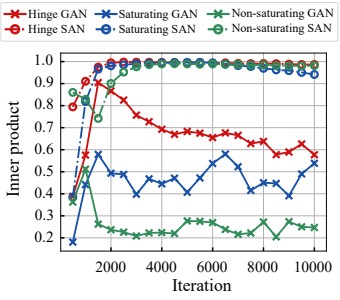

Figure 5: Comparison of the learned distributions (at 10,000 iterations) between GAN and SAN with various objectives. In all cases, SANs cover all modes whereas mode collapse occurs in some GAN cases.

Figure 6: Inner product of trained $\omega$ and numerically estimated *optimal direction* for $\tilde{\mu}_0^{r_{\mathcal{J}}}$ and $\tilde{\mu}_\theta^{r_{\mathcal{J}}}$ during training. The trained $\omega$ were closer to the *optimal direction* with SAN than with GAN.

$\mathcal{L}^h$, while $\mathcal{L}^\omega$ enforces *direction optimality* on $\omega$. Hence, we can convert most GAN maximization problems into SAN problems by using Eq. (15) (also see Algorithm 1). One can also easily extend SAN to class conditional generation by introducing directions for respective classes (see Appx. C).

A key strength of SAN is that the model can be trained with just two small modifications to existing GAN implementations (see Fig. 4 and Appx. B). The first is to implement the discriminator's final linear layer to be on a hypersphere, i.e., $f_\varphi(x) = w_{\varphi_L}^\top h_{\varphi_{<L}}(x)$ with $w_{\varphi_L} \in \mathbb{S}^{D-1}$, where $\varphi_l$ is the parameter for the $l$th layer. The second is to use the maximization problem defined in Eq. (15).

**Comparison with other generative models.** First, we compare SAN with GAN. In GAN, other than Wasserstein GAN, the optimal discriminator is known to involve a dilemma: it induces a certain dissimilarity but leads to exploding and vanishing gradients (Arjovsky et al., 2017; Lin et al., 2021). In contrast, in SAN, the *metrizable conditions* ensure that the optimal discriminator is not necessary to obtain a *metrizable* discriminator. Next, similarly to variants of the maximum SW (Deshpande et al., 2019; Kolouri et al., 2019), SAN's discriminator is interpreted as extracting features $h(x)$ and slicing them in the most distinctive direction $\omega$ as in Fig. 1. In the conventional methods, one-dimensional Wasserstein distances are approximated by sample sorting (see Remark 4.2). It has been reported that the generation performances get better when the batch size increases (Nguyen et al., 2021; Chen et al., 2022), but larger batch sizes lead to slower training speeds, and the size is capped by memory constraints. In contrast, SAN removes the necessity of the approximation.

# 7 EXPERIMENTS

We perform experiments with synthetic and image datasets to (1) verify our perspective on GANs as presented in Sec. 5 in terms of *direction optimality*, *separability*, and *injectivity*, and (2) show the effectiveness of SAN against GAN. For fair comparisons, we essentially use the same architectures in SAN and GAN. However, we modify the last linear layer of SAN's discriminators (see Sec. 6).

## 7.1 MIXTURE OF GAUSSIANS

To empirically investigate *optimal direction*, we conduct experiments on a mixture of Gaussian (MoG). Please refer to Appx. H for empirical verification of the implications of Theorem 5.3 regarding *separability* and *injectivity*. We use a two-dimensional sample space $X = \mathbb{R}^2$. The target MoG on $X$ comprises eight isotropic Gaussians with variances $0.05^2$ and means distributed evenly on a circle of radius $1.0$. We utilize a 10-dimensional latent space $Z$ to model a generator measure.

Table 3: FID scores (↓) on DCGAN.

| Dataset | Hinge loss | | Saturating loss | | Non-saturating loss | |
|---------|------------|--|-----------------|--|---------------------|--|
| | GAN | SAN | GAN | SAN | GAN | SAN |
| CIFAR10 | $24.07_{\pm0.56}$ | $\textbf{20.23}_{\pm0.86}$ | $25.63_{\pm0.98}$ | $\textbf{20.62}_{\pm0.94}$ | $24.90_{\pm0.21}$ | $\textbf{20.51}_{\pm0.36}$ |
| CelebA | $32.51_{\pm2.53}$ | $\textbf{27.79}_{\pm1.60}$ | $37.33_{\pm1.02}$ | $\textbf{28.16}_{\pm1.60}$ | $28.22_{\pm2.16}$ | $\textbf{27.78}_{\pm4.59}$ |

Table 4: FID and IS results with the experimental setup of BigGAN (Brock et al., 2019). Scores marked with ∗ are results from our implementation, which is based on the BigGAN authors' PyTorch implementation. For reference, scores reported in their paper are denoted with †.

| Metric | CIFAR10 | | | CIFAR100 | |
|--------|---------|--|--|----------|--|
| | Hinge GAN$^\dagger$ | Hinge GAN$^*$ | Hinge SAN$^*$ | Hinge GAN$^*$ | Hinge SAN$^*$ |
| FID (↓) | 14.73 | $8.25_{\pm0.82}$ | $\textbf{6.20}_{\pm0.27}$ | $10.73_{\pm0.16}$ | $\textbf{8.05}_{\pm0.04}$ |
| IS (↑) | 9.22 | $9.05_{\pm0.05}$ | $\textbf{9.16}_{\pm0.08}$ | $10.56_{\pm0.01}$ | $\textbf{10.72}_{\pm0.05}$ |

Table 5: Numerical results for StyleGAN-XL (Sauer et al., 2022) and StyleSAN-XL. Scores marked with † and ‡ are reported in their paper and repository, respectively. Note that our StyleSAN-XL model trained on CIFAR10 is larger in model size than StyleGAN-XL.

| Method | CIFAR10 | | ImageNet (256×256) | |
|--------|---------|--|--------------------|--|
| | StyleGAN-XL$^\ddagger$ | StyleSAN-XL$^*$ | StyleGAN-XL$^\dagger$ | StyleSAN-XL$^*$ |
| FID (↓) | (1.85) | **1.36** | 2.30 | **2.14** |
| IS (↑) | – | – | 265.12 | **274.20** |

We compare SAN and GAN with various objectives. As shown in Fig. 5, the generator measures trained with SAN cover all modes, whereas mode collapse (Srivastava et al., 2017) occurs with hinge GAN and non-saturating GAN. In addition, Fig. 6 shows a plot of the inner product of the learned direction $\omega$ (or the normalized weight in GAN's last linear layer) and the estimated *optimal direction* for $\tilde{\mu}_0^{r_{\mathcal{J}}}$ and $\tilde{\mu}_\theta^{r_{\mathcal{J}}}$. Recall that there is no guarantee that a non-optimal direction $\omega$ induces a distance.

## 7.2 IMAGE GENERATION

We apply the SAN scheme to image generation tasks to show it scales beyond toy experiments.

**DCGAN.** We train SANs and GANs with various objective functions on CIFAR10 (Krizhevsky et al., 2009) and CelebA (128×128) (Liu et al., 2015). We adopt the DCGAN architectures (Radford et al., 2016), and for the discriminator, we apply spectral normalization (Miyato et al., 2018). As shown in Table 3, SANs outperform GANs in terms of the FID score in all cases.

**BigGAN.** Next, we apply SAN to BigGAN (Brock et al., 2019) on CIFAR10 and CIFAR100. In this experiment, we calculate the Inception Score (IS) (Salimans et al., 2016), as well as the FID, to evaluate the sample quality by following the experiment in the original paper. As in Table 4, the adoption of SAN consistently improves the generation performance in terms of both metrics.

**StyleGAN-XL.** Lastly, we apply our SAN training framework to StyleGAN-XL (Sauer et al., 2022), which is a state-of-the-art GAN-based generative model. We name the StyleGAN-XL model combined with our SAN training framework StyleSAN-XL. We train StyleSAN-XL on CIFAR10 and ImageNet (256×256) (Russakovsky et al., 2015), and report the numerical results, as with BigGAN. As shown in Table 5, our models improve the state-of-the-art scores on both cases.

## 8 CONCLUSION

We have proposed a unique perspective on GANs to derive sufficient conditions for the discriminator to serve as a distance between the data and generator probability measures. To this end, we introduced the FM$^*$ and max-ASW families. By using a class of metrics that are included in both families, we derived the *metrizable conditions* for Wasserstein GAN. We then extended the result to a general GAN. The derived conditions consist of *direction optimality*, *separability*, and *injectivity*.

We then leveraged the theoretical results to develop the Slicing Adversarial Network (SAN), in which a generator and discriminator are trained with a modified GAN training scheme. Despite the ease of modifications and the generality, this model can impose *direction optimality* on the discriminator. Our experiments on synthetic and image datasets showed that SANs outperformed GANs in terms of both the sample quality and mode coverage.

## REPRODUCIBILITY STATEMENT

The experimental setups are provided in Appx. F with a detailed description of the network architectures and hyperparameters. We further provide our source code to reproduce our results at `https://github.com/sony/san`. Our implementation for all the image generation tasks is based on open-source repositories. The proofs of all the theoretical claims can be found in Appx. D.

## ACKNOWLEDGEMENTS

We sincerely acknowledge the support of Basavaraj Murali, Riya Kuriakose, Sonal Monteiro, Srinidhi Srinivasa, and Takuya Narihira, who assisted with our experiments during the rebuttal period. Computational resource of AI Bridging Cloud Infrastructure (ABCI) provided by National Institute of Advanced Industrial Science and Technology (AIST) was used.

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

# Contents

## A   NOTATIONS

We summarize several notations that are used throughout the paper but not introduced in Sec. 2.1 in Table 6.

## B   CONVERTING GANs TO SANs

We can convert GANs to SANs simply by adding two modifications to existing GAN implementations, as shown in Fig. 4. First, we propose to modify the last linear layer of the discriminator as in the following PyTorch-like code snippet. In the code snippet, `feature_network` represents any feature extractor $h(\cdot)$ built with nonlinear layers. On top of it, we propose using the novel maximization objective functions $\mathcal{L}^h$ and $\mathcal{L}^\omega$. The discriminator's outputs, `out_fun` and `out_dir`, are utilized for $\mathcal{L}^h$ and $\mathcal{L}^\omega$, respectively.

```python
import torch.nn as nn

class GANDiscriminator(nn.Module):
    def __init__(self,
        dim # dimension of output of "h(x)"
    ):
        super(GANDiscriminator, self).__init__()

        self.main = feature_network(dim) # nonlinear function "h(x)"
        self.fc = nn.Linear(dim, 1) # last linear layer "w"

    def forward(self,
        x # batch of real/fake samples
    ):
```

```
15        feature = self.main(x)
16        out = self.fc(feature)
17
18        return out
```

Listing 1: GAN discriminator

```
1  import torch
2  import torch.nn as nn
3  import torch.nn.functional as F
4
5  class SANDiscriminator(nn.Module):
6      def __init__(self,
7          dim # dimension of output of "h(x)"
8      ):
9          super(SANDiscriminator, self).__init__()
10
11          self.main = feature_network(dim) # nonlinear function "h(x)"
12          vec_init = torch.randn(dim, ) # initialization for direction
13          self.vec = nn.Parameter(vec_init) # unnormalized direction
14
15      def forward(self,
16          x # batch of real/fake samples
17      ):
18          feature = self.main(x)
19          direction = F.normalize(self.vec, dim=0) # direction "omega"
20          out_fun = feature @ direction.detach()
21          out_dir = feature.detach() @ direction
22
23          return out_fun, out_dir
```

Listing 2: SAN discriminator

As an example, converting the maximization objective of Hinge GAN to SAN leads to

$$\mathcal{V}_{\text{Hinge}}^{\text{SAN}}(\omega, h; \mu_\theta) = \underbrace{\mathcal{V}_{\text{Hinge}}(\langle\omega^-, h\rangle, \mu_\theta)}_{\mathcal{L}^h} + \lambda \cdot \underbrace{\mathcal{V}_{\text{W}}(\langle\omega, h^-\rangle, \mu_\theta)}_{\mathcal{L}^\omega}. \quad (16)$$

For general GANs, the approximations of $\tilde{\mu}_0^{r_{\mathcal{J}} \circ f}$ and $\tilde{\mu}_\theta^{r_{\mathcal{J}} \circ f}$ are needed. The objective term $\mathcal{L}^\omega$ can be formulated as

$$\mathcal{L}^\omega(\omega; h, \mu_\theta) = \frac{1}{Z_0}\mathbb{E}_{x \sim \mu_0}[r_{\mathcal{J}} \circ f^-(x)\langle\omega, h(x)\rangle] - \frac{1}{Z_\theta}\mathbb{E}_{x \sim \mu_\theta}[r_{\mathcal{J}} \circ f^-(x)\langle\omega, h(x)\rangle], \quad (17)$$

where $Z_0 = \mathbb{E}_{x \sim \mu_0}[r_{\mathcal{J}} \circ f^-(x)]$ and $Z_\theta = \mathbb{E}_{x \sim \mu_\theta}[r_{\mathcal{J}} \circ f^-(x)]$. The normalizing terms $Z_0$ and $Z_\theta$ can be approximated per batch. However, we found it beneficial to use the following auxiliary objective function instead:

$$\mathcal{L}_{\text{simple}}^\omega(\omega; h, \mu_\theta) = \mathbb{E}_{x \sim \mu_0}[R_{\mathcal{J}}(\langle\omega, h\rangle)] - \mathbb{E}_{x \sim \mu_\theta}[R_{\mathcal{J}}(\langle\omega, h\rangle)], \quad (18)$$

which ignores the normalization terms $Z_0$ and $Z_\theta$ in Eq. (17). The psuedo code can found in Algorithm1

## C  CLASS CONDITIONAL SAN

SAN can easily be extended to conditional generation. Let $C$ be the number of classes. For a conditional case, we adopt the usual generator parameterization, $g_\theta : Z \times \{c\}_{c=1}^C \to X$, which gives a conditional generator distribution $\mu_\theta^c := g_\theta(\cdot, c)_\sharp \sigma$. For the discriminator, we prepare trainable directions for the respective classes as $\Omega := \{\omega_c\}_{c=1}^C$. Under this setup, we define the minimax problem of conditional SAN as follows:

$$\max_{\Omega,h} \mathbb{E}_{c \sim P(c)} \left[\mathcal{V}_{\mu_0^c}^{\text{SAN}}(\omega_c, h; \mu_\theta^c)\right] \quad \text{and} \quad \min_\theta \mathbb{E}_{c \sim P(c)} \left[\mathcal{J}(\theta; \langle\omega_c, h\rangle)\right], \quad (19)$$

Table 6: Several notations used throughout the main paper.

| **Sliced optimal transport** | |
| --- | --- |
| $\mathcal{R}I_\mu(\cdot, \omega)$ | The RT of a probability density function $I$ with a direction $\omega \in \mathbb{S}^{D-1}$ given. |
| $\mathcal{S}^h I_\mu(\cdot, \omega)$ | The SRT of a probability density function $I$ with a direction $\omega \in \mathbb{S}^{D-1}$ and function $h : X \to \mathbb{R}^D$ given. |
| $F_\mu^{h,\omega}(\cdot)$ | The cumulative distribution function of $\mathcal{S}^h I_\mu(\cdot, \omega)$. |
| **Functional spaces** | |
| $\mathcal{F}(X)$ | Any functional space included in $L^\infty(X, \mathbb{R}^D)$. |
| $\mathcal{F}_{\mathrm{S}(\mu,\nu)}(X)$ | The class of all the separable functions for $\mu, \nu \in \mathcal{P}(X)$, which is often simply denoted as $\mathcal{F}_\mathrm{S}$. |
| $\mathcal{F}_\mathrm{I}(X)$ | The class of all the injectivity functions in $L^\infty(X, \mathbb{R}^D)$, which is often simply denoted as $\mathcal{F}_\mathrm{I}$. |
| $\mathcal{F}_{\mathrm{I},\mathrm{S}}(X)$ | The intersection set of $\mathcal{F}_\mathrm{I}$ and $\mathcal{F}_\mathrm{S}$, i.e., $\mathcal{F}_{\mathrm{I},\mathrm{S}} = \mathcal{F}_\mathrm{I} \cap \mathcal{F}_\mathrm{S}$. |
| **Divergences and distances** | |
| $\mathscr{D}_D^\mathrm{FM}$ | The family of FM with $D \in \mathbb{N}$ given. Its instance is denoted as $FM_\mathcal{F}(\mu, \nu) \in \mathscr{D}_D^\mathrm{FM}$. |
| $\mathscr{D}_\mathcal{F}^{\mathrm{FM}^*}$ | The family of FM$^*$ with $\mathcal{F}$ given. Its instance is denoted as $FM_h^*(\mu, \nu) \in \mathscr{D}_\mathcal{F}^{\mathrm{FM}^*}$ with $h \in \mathcal{F}$. |
| $\mathscr{D}_\mathcal{F}^\mathrm{mA}$ | The family of max-ASW with $\mathcal{F}$ given. Its instance is denoted as $max\text{-}ASW_h(\mu, \nu) \in \mathscr{D}_\mathcal{F}^\mathrm{mA}$ with $h \in \mathcal{F}$. |

---

**Algorithm 1** Training SAN (the blue lines indicate modified steps against GAN training)

---

**Require:** Consider a sample space $X$ and a latent space $Z$. A target distribution $\mu_0 \in \mathcal{P}(X)$ and a base distribution $\sigma \in \mathcal{P}(Z)$ are given. Initialize a parameter $\theta$ modeling a generator function as $g_\theta : Z \to X$, and discriminator parameters, $\varphi$ and $\omega \in \mathbb{S}^{D-1}$, modeling a discriminator function as $\langle \omega, h_\varphi \rangle$, where $h_\varphi : X \to \mathbb{R}^D$. Step sizes for the trainable parameters are set as $\eta_\theta$, $\eta_\varphi$, and $\eta_\omega$. Original minimization and maximization problems for GANs, denoted as $\mathcal{J}_\mathrm{GAN}$ and $\mathcal{V}_\mathrm{GAN}$, are given.
**while** not converged **do**
    /∗ Discriminator update ∗/
    **for** certain number of steps **do**
        Sample a minibatch of data $\{x_i\}_{i=1}^M \sim \mu_0$ (i.i.d.).
        Get generated samples $\{y_i\}_{i=1}^N$, where $y_i = g_\theta(z_i)$ with $\{z_i\}_{i=1}^N \sim \sigma$ (i.i.d.).
        Compute $\hat{\mathcal{L}}^h$ as the Monte-Carlo estimate of $\mathcal{V}_\mathrm{GAN}(\langle \omega, h \rangle; \theta)$ with $\{x_i\}_{i=1}^M$ and $\{y_i\}_{i=1}^N$.
        Compute $\hat{\mathcal{L}}^\omega$ as the Monte-Carlo estimate of Eq. (18) with $\{x_i\}_{i=1}^M$ and $\{y_i\}_{i=1}^N$.
        Update the parameters as $\omega \leftarrow \mathrm{Proj}_{\mathbb{S}^{D-1}}(\omega + \nabla_\omega \hat{\mathcal{L}}^\omega)$ and $\varphi \leftarrow \varphi + \eta_\varphi \nabla_\varphi \hat{\mathcal{L}}^\varphi$.
    **end for**
    /∗ Generator update ∗/
    **for** certain number of steps **do**
        Get generated samples $\{y_i\}_{i=1}^N$, where $y_i = g_\theta(z_i)$ with $\{z_i\}_{i=1}^N \sim \sigma$ (i.i.d.).
        Compute $\tilde{\mathcal{J}}$ as the Monte-Carlo estimate of $\mathcal{J}_\mathrm{GAN}(\theta, \langle \omega, h \rangle)$ with $\{y_i\}_{i=1}^N$.
        Update the parameters as $\theta \leftarrow \theta - \eta_\theta \nabla_\theta \tilde{\mathcal{J}}$.
    **end for**
**end while**

---

where $P(c)$ is a pre-fix probability mass function for $c$, and $\mu_0^c$ is the target probability measure conditioned on $c$.

GAN can easily be converted into SAN even for conditional generation, as many previous works have implemented a conditional discriminator with the class-conditional last layer as $w_{\phi_L,c}^\top h_{\phi_{<L}}(x)$ (Miyato & Koyama, 2018; Miyato et al., 2018; Brock et al., 2019; Zhang et al., 2019).

## D    PROOFS

**Proposition 3.2.**  *For $\mathcal{F}(X) \in L^\infty(X, \mathbb{R})$, $IPM_\mathcal{F}(\cdot, \cdot) := \max_{f \in \mathcal{F}} d_f(\cdot, \cdot) \in \mathscr{D}_1^{FM}$.*

*Proof.*  The claim is a direct consequence of the definitions of the IPM and the FM.    □

We introduce a lemma, which will come in handy for the proofs of Proposition 3.5 and Lemma 4.3.

**Lemma D.1.** *Given a function $h \in L^\infty(X, \mathbb{R}^D)$ and probability measures $\mu, \nu \in \mathcal{P}(X)$, we have*

$$\|d_h(\mu, \nu)\|_2 = \max_{\omega \in \mathbb{S}^{D-1}} d_{\langle \omega, h \rangle}(\mu, \nu). \tag{20}$$

*Proof.* From the Cauchy–Schwarz inequality, we have the following lower bound using $\omega \in \mathbb{S}^{D-1}$:

$$\begin{aligned}
\|d_h(\mu, \nu)\|_2 &= \|\mathbb{E}_{x \sim \mu}[h(x)] - \mathbb{E}_{x \sim \nu}[h(x)]\|_2 \\
&\geq \langle \omega, \mathbb{E}_{x \sim \mu}[h(x)] - \mathbb{E}_{x \sim \nu}[h(x)] \rangle,
\end{aligned} \tag{21}$$

where the equality holds if and only if $\omega = \hat{d}(\mu, \nu)$. Recall that $\hat{(\cdot)}$ denotes a normalization operator. Then, calculating the norm $\|d_h(\mu, \nu)\|_2$ is formulated as the following maximization problem:

$$\|d_h(\mu, \nu)\|_2 = \max_{\omega \in \mathbb{S}^{D-1}} \mathbb{E}_{x \sim \mu}[\langle \omega, h(x) \rangle] - \mathbb{E}_{x \sim \nu}[\langle \omega, h(x) \rangle] \tag{22}$$

$$= \max_{\omega \in \mathbb{S}^{D-1}} d_{\langle \omega, h \rangle}(\mu, \nu). \tag{23}$$

Note that we can interchange the inner product and the expectation operator by the independency of $\omega$ w.r.t. $x$. $\qquad\square$

**Proposition 3.5.** *Let $\omega$ be on $\mathbb{S}^{D-1}$. For any $h \in L^\infty(X, \mathbb{R}^D)$, minimization of $FM_h^*(\mu_\theta, \mu_0)$ is equivalent to optimization of $\min_{\theta \in \mathbb{R}^{D_\theta}} \max_{\omega \in \mathbb{S}^{D-1}} \mathcal{J}_W(\theta; \langle \omega, h \rangle)$. Thus,*

$$\nabla_\theta FM_h^*(\mu_\theta, \mu_0) = \nabla_\theta \mathcal{J}_W(\theta; \langle \omega^*, h \rangle), \tag{24}$$

*where $\omega^*$ is the optimal solution (direction) given as follows:*

$$\omega^* = \arg\max_{\omega \in \mathbb{S}^{D-1}} d_{\langle \omega, h \rangle}(\mu_0, \mu_\theta). \tag{25}$$

*Proof.* From Lemma D.1, we have

$$\|d_h(\mu_0, \mu_\theta)\|_2 = \max_{\omega \in \mathbb{S}^{D-1}} d_{\langle \omega, h \rangle}(\mu_0, \mu_\theta) \tag{26}$$

$$= \max_{\omega \in \mathbb{S}^{D-1}} \mathbb{E}_{x \sim \mu_0}[\langle \omega, h(x) \rangle] - \mathbb{E}_{x \sim \mu_\theta}[\langle \omega, h(x) \rangle]. \tag{27}$$

By a simple envelope theorem (Milgrom & Segal, 2002), we obtain

$$\nabla_\theta \|d_h(\mu_0, \mu_\theta)\|_2 = \nabla_\theta \left( \mathbb{E}_{x \sim \mu_0}[\langle \omega^*, h(x) \rangle] - \mathbb{E}_{x \sim \mu_\theta}[\langle \omega^*, h(x) \rangle] \right), \tag{28}$$

where $\omega^* = \max_{\omega \in \mathbb{S}^{D-1}} d_{\langle \omega, h \rangle}(\mu_0, \mu_\theta)$. Since the first term in the right-hand side of Eq. (28) does not depend on $\theta$, taking the gradient of $\|d_h(\mu_0, \mu_\theta)\|_2$ w.r.t. $\theta$ becomes

$$\nabla_\theta \|d_h(\mu_0, \mu_\theta)\|_2 = -\nabla_\theta \mathbb{E}_{x \sim \mu_\theta}[\langle \omega^*, h(x) \rangle]. \tag{29}$$

Substituting the definition of $\mathcal{J}_W$ and $FM_h^*(\cdot, \cdot)$ into Eq. (29) leads to the equality in the statement of Proposition 3.5. $\qquad\square$

**Lemma 4.3.** *Given $\mu, \nu \in \mathcal{P}(X)$, every $h \in \mathcal{F}_{S(\mu,\nu)}(X)$ satisfies $FM_h^*(\mu, \nu) \in \mathscr{D}_{\mathcal{F}_S}^{mA}$.*

*Proof.* From Lemma D.1, we have

$$\|d_h(\mu, \nu)\|_2 = \max_{\omega \in \mathbb{R}^{D-1}} \mathbb{E}_{x \sim \mu}[\langle \omega, h(x) \rangle] - \mathbb{E}_{x \sim \nu}[\langle \omega, h(x) \rangle] \tag{30}$$

$$= \mathbb{E}_{x \sim \mu}[\langle \omega^*, h(x) \rangle] - \mathbb{E}_{x \sim \nu}[\langle \omega^*, h(x) \rangle], \tag{31}$$

where the maximizer is $\omega^* = \hat{d}_h(\mu, \nu)$.

The expected values are reformulated by the SRT as

$$\mathbb{E}_{x\sim\mu}[\langle\omega, h(x)\rangle] = \int_X \left(\int_{\mathbb{R}} \xi\delta(\xi - \langle\omega, h(x)\rangle)d\xi\right) I_\mu(x)dx \tag{32}$$

$$= \int_{\mathbb{R}} \xi\left(\int_X I_\mu(x)\delta(\xi - \langle\omega, h(x)\rangle)dx\right) d\xi \tag{33}$$

$$= \mathbb{E}_{\xi\sim\mathcal{S}^h I_\mu(\xi,\omega)}[\xi]. \tag{34}$$

The interchange of the integral follows from Fubini's theorem. Here, the applicability of the theorem is justified by the absolute integrability of the integrant function. That is, $\int_X \|h(x)\|_2 d\mu(x) \leq \|h\|_{L^\infty(X,\mathbb{R}^D)}\mu(X) = \|h\|_{L^\infty(X,\mathbb{R}^D)} < \infty$.

Further, the right-hand side of Eq. (34) follows the representation of an expectation by an integral of quantiles:

$$\mathbb{E}_{\xi\sim\mathcal{S}^h I_\mu(\xi,\omega)}[\xi] = \int_0^1 Q_\mu^{h,\omega}(\rho)d\rho \tag{35}$$

By substituting Eqs. (34) and (35) into Eq. (31), we obtain

$$\|d_h(\mu,\nu)\|_2 = \int_0^1 \left(Q_\mu^{h,\omega^*}(\rho) - Q_\nu^{h,\omega^*}(\rho)\right) d\rho. \tag{36}$$

The right-hand side of Eq. (36) is bounded above:

$$\|d_h(\mu,\nu)\|_2 \leq \int_0^1 \left|Q_\mu^{h,\omega^*}(\rho) - Q_\nu^{h,\omega^*}(\rho)\right| d\rho. \tag{37}$$

The equality holds if and only if $Q_\mu^{h,\omega^*}(\rho) - Q_\nu^{h,\omega^*}(\rho) \geq 0$ for all $\rho$ or $Q_\mu^{h,\omega^*}(\rho) - Q_\nu^{h,\omega^*}(\rho) \leq 0$ for all $\rho$. By considering the fact that selecting $-\omega$ changes the sign and Eq. (36) is non-negative, the necessary and sufficient condition for the equality becomes $Q_\mu^{h,\omega^*}(\rho) - Q_\nu^{h,\omega^*}(\rho) \geq 0$ for $\rho \in [0, 1]$. From the monotonicity of quantile functions, the condition for the equality is written as $F_\mu^{h,\omega}(\rho) \leq F_\nu^{h,\omega}(\rho)$. $\square$

We here borrow a lemma from Chen et al. (2022), which is used in the proof of Lemma 4.4.

**Lemma D.2** (Chen et al. (2022), Lemma 1). *Given an injective function $h : X \to \mathbb{R}^D$ and two probability measures $\mu, \nu \in \mathcal{P}(X)$, for all $\xi \in \mathbb{R}$ and $\omega \in \mathbb{S}^{D-1}$, $\mathcal{S}^h I_\mu(\xi,\omega) = \mathcal{S}^h I_\nu(\xi,\omega)$ if and only if $\mu = \nu$.*

**Lemma 4.4.** *Every max-ASW$_h(\cdot,\cdot) \in \mathscr{D}_{\mathcal{F}_I}^{mA}$ is a distance, where $\mathcal{F}_I$ indicates a class of injective functions in $L^\infty(X,\mathbb{R}^D)$.*

*Proof.* One can prove this claim by following similar procedures to the proof of Proposition 1 in Kolouri et al. (2019).

First, the non-negativity and symmetry properties are immediately proven by the fact that the Wasserstein distance is indeed a distance satisfying non-negatibity and symmetry.

Next, let $\omega^*$ denote

$$\omega^* = \arg\max_{\omega\in\mathbb{S}^{D-1}} W_1(\mathcal{S}^h I_{\mu_1}(\cdot,\omega), \mathcal{S}^h I_{\mu_2}(\cdot,\omega)). \tag{38}$$

Pick $\nu \in \mathcal{P}(X)$ arbitrarily. The triangle inequality with *max-ASW$_h(\cdot,\cdot)$* is satisfied since

$$max\text{-}ASW_h(\mu_1,\mu_2) = \max_{\omega\in\mathbb{S}^{D-1}} W_1(\mathcal{S}^h I_{\mu_1}(\cdot,\omega), \mathcal{S}^h I_{\mu_2}(\cdot,\omega))$$

$$= W_1(\mathcal{S}^h I_{\mu_1}(\cdot,\omega^*), \mathcal{S}^h I_{\mu_2}(\cdot,\omega^*))$$

$$\leq W_1(\mathcal{S}^h I_{\mu_1}(\cdot,\omega^*), \mathcal{S}^h I_\nu(\cdot,\omega^*)) + W_1(\mathcal{S}^h I_\nu(\cdot,\omega^*), \mathcal{S}^h I_{\mu_2}(\cdot,\omega^*)) \tag{39}$$

$$\leq \max_{\omega\in\mathbb{R}^{D-1}} W_1(\mathcal{S}^h I_{\mu_1}(\cdot,\omega), \mathcal{S}^h I_\nu(\cdot,\omega^*))$$

$$+ \max_{\omega\in\mathbb{R}^{D-1}} W_1(\mathcal{S}^h I_\nu(\cdot,\omega^*), \mathcal{S}^h I_{\mu_2}(\cdot,\omega)) \tag{40}$$

$$= max\text{-}ASW_h(\mu_1,\nu) + max\text{-}ASW_h(\nu,\mu_2). \tag{41}$$

We next show the identity of indiscernibles. For any $\mu$, from $W_1(\mu, \mu) = 0$, we have $W_1(\mathcal{S}^h I_\mu(\cdot, \omega), \mathcal{S}^h I_\mu(\cdot, \omega)) = 0$; therefore, $\textit{max-ASW}_h(\mu, \mu) = 0$. Conversely, $\textit{max-ASW}_h(\mu, \nu) = 0$ is equivalent to $\mathcal{S}^h I_\mu(\cdot, \omega) = \mathcal{S}^h I_\nu(\cdot, \omega)$ for any $\omega \in \mathbb{S}^{D-1}$. This holds if and only if $\mu = \nu$ from the injectivity assumption on $h$ and Lemma D.2. Therefore, $\textit{max-ASW}_h(\mu, \nu) = 0$ holds if and only if $\mu = \nu$. $\qquad\square$

**Proposition 4.5.** *Given $\mu, \nu \in \mathcal{P}(X)$, every $FM_h^*(\mu, \nu) \in \mathscr{D}_{\mathcal{F}_{I,S}}^{FM^*}$ is indeed a distance, where $\mathcal{F}_{I,S}$ indicates $\mathcal{F}_I \cap \mathcal{F}_S$.*

*Proof.* The claim is a direct consequence of the results of Lemmas 4.3 and 4.4.

From $\mathcal{F}_{I,S} := \mathcal{F}_I \cap \mathcal{F}_S \subset \mathcal{F}_S$ and Lemma 4.3, $\mathcal{D}(\mu, \nu) \in \mathscr{D}_{\mathcal{F}_{I,S}}^{\text{mA}}$ holds for any $\mathcal{D}(\mu, \nu) \in \mathscr{D}_{\mathcal{F}_{I,S}}^{FM^*}$. At the same time, from $\mathcal{F}_{I,S} \subset \mathcal{F}_I$ and Lemma 4.4, $\mathcal{D}(\mu, \nu) \in \mathscr{D}_{\mathcal{F}_{I,S}}^{\text{mA}}$ is a distance. Thus, the claim is proven. $\qquad\square$

**Lemma 5.1.** *Given $h \in \mathcal{F}_I \cap \mathcal{F}_S$ with $\mathcal{F}_I, \mathcal{F}_S \subseteq L^\infty(X, \mathbb{R}^D)$, let $\omega^* \in \mathbb{S}^{D-1}$ be $\hat{d}_h(\mu_0, \mu_\theta)$. Then $f(x) = \langle \omega, h(x) \rangle$ is $(\mathcal{J}_W, FM_h^*)$-metrizable.*

*Proof.* From Proposition 4.5, given that $h$ is injective and separable for $\mu_0$ and $\mu_\theta$, $FM_h^*(\mu_0, \mu_\theta)$ is a distance between $\mu_0$ and $\mu_\theta$.

From Proposition 3.5, minimizing $FM_h^*(\mu_0, \mu_\theta)$ w.r.t. $\theta$ is equivalent to optimizing the following:

$$\min_\theta \mathcal{J}_W(\theta, \langle \omega^*, h \rangle), \tag{42}$$

where $\omega^*$ is the maximizer of $\hat{d}_{\langle \omega, h \rangle}(\mu_0, \mu_\theta)$, i.e., $\omega^* = d_h(\mu_0, \mu_\theta)$. Now, $FM_h^*(\mu_0, \mu_\theta)$ is distance, so $x \mapsto \langle \omega^*, h(x) \rangle$ is $(\mathcal{J}_W, FM_h^*)$-*metrizable* for $\mu_0$ and $\mu_\theta$. $\qquad\square$

**Lemma 5.2.** *For any $\rho : X \to \mathbb{R}_+$ and a distance for probability measures $\mathcal{D}(\cdot, \cdot)$, $\mathcal{D}(\tilde{\mu}^\rho, \tilde{\nu}^\rho)$ indicates a distance between $\mu$ and $\nu$.*

*Proof.* For notational simplicity, we denote $\mathcal{D}(\tilde{\mu}^\rho, \tilde{\nu}^\rho)$ as $\tilde{\mathcal{D}}_\rho(\mu, \nu)$. First, the non-negativity and symmetry properties are immediately proven by the assumption that $\mathcal{D}(\cdot, \cdot)$ is a distance.

The triangle inequality regarding $\tilde{\mathcal{D}}_\rho(\cdot, \cdot)$ is satisfied since the following inequality holds for any $\mu_1, \mu_2, \mu_3 \in \mathcal{P}(X)$:

$$\begin{aligned} \tilde{\mathcal{D}}_\rho(\mu_1, \mu_2) &= \mathcal{D}(\tilde{\mu}_1^\rho, \tilde{\mu}_2^\rho) \\ &\leq \mathcal{D}(\tilde{\mu}_1^\rho, \tilde{\mu}_3^\rho) + \mathcal{D}(\tilde{\mu}_3^\rho, \tilde{\mu}_2^\rho) \end{aligned} \tag{43}$$

where $I_{\tilde{\mu}_i^\rho}(x) \propto \rho(x) I_{\mu_i}(x)$ for $i = 1, 2, 3$. Now, from the positivity assumption on $\rho$, $I_\mu \mapsto I_{\tilde{\mu}^\rho}$ is invertible. In particular, one can recover $\mu_i$ from $\tilde{\mu}_i^\rho$ uniquely by

$$I_{\mu_i}(x) = \frac{I_{\tilde{\mu}_i^\rho}(x)}{\rho(x) \int_X \rho(x)^{-1} d\mu_i(x)}. \tag{44}$$

Thus, with the $\mu_i$ given by Eq. (44), we have

$$\tilde{\mathcal{D}}_\rho(\mu_i, \mu_3) = \mathcal{D}(\tilde{\mu}_i^\rho, \tilde{\mu}_3^\rho) \quad (i = 1, 2). \tag{45}$$

From Eqs. (43) and (45), we have

$$\tilde{\mathcal{D}}_\rho(\mu_1, \mu_2) \leq \tilde{\mathcal{D}}_\rho(\mu_1, \mu_3) + \tilde{\mathcal{D}}_\rho(\mu_3, \mu_2). \tag{46}$$

Obviously, $\tilde{\mathcal{D}}_\rho(\mu, \mu) = 0$. Now, $\tilde{\mathcal{D}}_\rho(\mu, \nu) = 0$ is equivalent to $\tilde{\mu}^\rho = \tilde{\nu}^\rho$. Here, from the invertibility of $I_\mu \mapsto I_{\tilde{\mu}^\rho}$ again, $\tilde{\mu}^\rho = \tilde{\nu}^\rho$ if and only if $\mu = \nu$. Thus, $\tilde{\mathcal{D}}_\rho(\mu, \nu) = 0 \iff \mu = \nu$. $\qquad\square$

**Theorem 5.3.** *Given a functional $\mathcal{J}(\theta; f) := \mathbb{E}_{x \sim \mu_\theta}[R \circ f(x)]$ with $R : \mathbb{R} \to \mathbb{R}$ whose derivative, denoted as $R'$, is positive, let $h \in L^\infty(X, \mathbb{R}^D)$ and $\omega \in \mathbb{S}^{D-1}$ satisfy the following conditions. Then, $f(x) = \langle \omega, h(x) \rangle$ is $\mathcal{J}$-metrizable for $\mu_\theta$ and $\mu_0$.*

- **(Direction optimality)** $\omega = \arg\max_{\tilde{\omega} \in \mathbb{S}^{D-1}} d_{\langle \tilde{\omega}, h \rangle}(\tilde{\mu}_0^{R' \circ f}, \tilde{\mu}_\theta^{R' \circ f})$,

- **(Separability)** $h$ is separable for $\tilde{\mu}_0^{R' \circ f}$ and $\tilde{\mu}_\theta^{R' \circ f}$,

- **(Injectivity)** $h$ is an injective function.

*Then, $x \mapsto \langle \omega, h(x) \rangle$ is metrizable with $\mathcal{J}$ for $\mu_\theta$ and $\mu_0$.*

*Proof.* For notational simplicity, we omit the superscripts of $\tilde{\mu}_0^{R' \circ f}$ and $\tilde{\mu}_\theta^{R' \circ f}$ in this proof. From Lemma 5.2, if $FM_h^*(\tilde{\mu}_0, \tilde{\mu}_\theta)$ is a distance between $\tilde{\mu}_0$ and $\tilde{\mu}_\theta$, then it is also a distance between $\mu_0$ and $\mu_\theta$.

Following the same procedures as in Lemma 5.1 leads to the claim.

From Proposition 4.5, given that $h$ is injective and separable for $\tilde{\mu}_0$ and $\tilde{\mu}_\theta$, $FM_h^*(\tilde{\mu}_0, \tilde{\mu}_\theta)$ is a distance between $\tilde{\mu}_0$ and $\tilde{\mu}_\theta$.

From Proposition 3.5, minimizing $FM_h^*(\tilde{\mu}_0, \tilde{\mu}_\theta)$ w.r.t. $\theta$ is equivalent to optimizing the following:

$$\min_\theta \tilde{\mathcal{J}}_W(\theta, \langle \omega^*, h \rangle), \tag{47}$$

where $\tilde{\mathcal{J}}_W(\theta, \langle \omega^*, h \rangle) = -\mathbb{E}_{x \sim \tilde{\mu}_\theta}[\langle \omega^*, h(x) \rangle]$ and $\omega^* = \arg\max_{\tilde{\omega} \in \mathbb{S}^{D-1}} d_{\langle \tilde{\omega}, h \rangle}(\tilde{\mu}_0^{R' \circ f}, \tilde{\mu}_\theta^{R' \circ f})$. Now that $\omega = \omega^*$ from the direction optimality assumption, the gradient of the minimization objective in Eq. (47) w.r.t. $\theta$ becomes

$$\nabla_\theta \tilde{\mathcal{J}}_W(\theta, \langle \omega^*, h \rangle) = -\nabla_\theta \mathbb{E}_{x \sim \tilde{\mu}_\theta}[\langle \omega, h(x) \rangle] \tag{48}$$

$$= -\frac{1}{Z} \mathbb{E}_{z \sim \sigma}[R' \circ f(g_\theta(z)) \nabla_\theta \langle \omega, h(g_\theta(z)) \rangle] \tag{49}$$

$$= \frac{1}{Z} \nabla_\theta \mathcal{J}(\theta, \langle \omega, h(g_\theta(z)) \rangle), \tag{50}$$

where we used an envelope theorem (Milgrom & Segal, 2002) to get Eq. (48) and $Z := \mathbb{E}_{x \sim \mu_\theta}[r \circ f(x)]$ is a normalizing constant.

Now, $FM_h^*(\tilde{\mu}_0, \tilde{\mu}_\theta)$ is distance and the gradient of $FM_h^*(\tilde{\mu}_0, \tilde{\mu}_\theta)$ is equivalent to $\nabla_\theta \mathcal{J}(\theta, \langle \omega, h(g_\theta(z)) \rangle)$ except for the scaling factor $Z$, so $x \mapsto \langle \omega, h(x) \rangle$ is $\mathcal{J}$-metrizable. $\qquad\square$

# E   RELATED WORK

In this section, we present the related work, some of which are described in Secs. 1, 2, and 6, while mentioning the limitations of the current work.

## E.1   LITERATURE ON GAN

Goodfellow et al. (2014) was responsible for pioneering work that developed GAN with various theoretical analyses under the optimal discriminator (maximizer) assumption. Since then, GAN has been applied to a variety of tasks, yet the instability of its learning has always been a challenge. Arjovsky et al. (2017) elaborated on one of the causes by showing that the maximizers of $f$-GANs lead to exploding and vanishing gradients with disjoint supports of the generator and data distributions. Although they proposed Wasserstein GAN, which avoids the gradient issue, it suffers from another instability issue (Chu et al., 2020; Xu et al., 2020; Qin et al., 2020). We empirically observe that Wasserstein GAN tends not to satisfy *separability*, which gives us another explanation for the instability. Several techniques to improve the GAN training for general GANs have been proposed, such as regularization terms (Nagarajan & Kolter, 2017; Terjék, 2020; Zhang et al., 2020), guidance losses (Sauer et al., 2022; 2023), and optimization schemes (Yaz et al., 2019; Chavdarova et al., 2019; Qin et al., 2020; Chavdarova et al., 2022). It should be noted that the above techniques are orthogonal to our proposed method imposing *direction optimality*, which means SAN has potential to enhance many existing GANs. Our experiment on StyleGAN-XL, which adopts many of the techniques to achieve the state-of-the-art performance in large-scale generation tasks (Sauer et al., 2022), supports this expectation (see Sec. 7.2).

Several papers have focused on a theoretical understanding of GAN optimization. The discriminator optimality is a convenient assumption for theoretical analysis (Goodfellow et al., 2014; Johnson & Zhang, 2019; Gao et al., 2019; Fan et al., 2022), although achieving such optimality in practical cases is rarely feasible (Li et al., 2018; Chu et al., 2020). Some studies shed light on the analysis of GAN optimization from the perspective of the local stability and convergence behavior of gradient descent-ascent (Nagarajan & Kolter, 2017; Mescheder et al., 2018). In another direction, researches have introduced and characterized variants of equilibrium notions such as the local equilibria (Ratliff et al., 2013; Jin et al., 2020), proximal equilibria (Farnia & Ozdaglar, 2020), and mixed Nash equilibria (Arora et al., 2017; Hsieh et al., 2019). Optimization smoothness, a useful perspective for looking into stability of gradient-based learning dynamics, is beneficial even for GAN cases under certain assumptions (Nitanda & Suzuki, 2018; Chu et al., 2020; Fiez et al., 2022). In our work, we analyze GAN optimization in terms of the concept of *metrizability* (see Definition 1.1), which does not fall into the above taxonomy.

### E.2 LITERATURE ON SLICED WASSERSTEIN DISTANCE

SW was initially proposed to mitigate the course of dimension (Bonneel et al., 2015). The calculation of SW involves Radon transform and direction sampling ($\omega$). Kolouri et al. (2019) first extended SW to the generalized SW in the context of deep learning. They replaced Radon transform with a generalized Radon transform (GRT) by using defining functions, which allows the use of a broad class of injective functions. Further, Chen et al. (2022) proposed the SRT to make it easier to incorporate neural functions into a variant of Radon transform. There are also some variants of Radon Transform, which take into account the properties of the target data geometry (Bonet et al., 2023) and neural layers (Nguyen & Ho, 2022).

Regarding the slicing part, the most basic approach is to sample directions according to uniform distribution on the hypersphere to discretely approximate the integral of Eq. (8). Since the random nature of slices could result in an underestimation of the distance (Kolouri et al., 2019), more efficient sampling schemes have been investigated. Deshpande et al. (2019) proposed the concept of max-SW, which uses only the direction that maximizes the one-dimensional Wasserstein distance (e.g., the integrand in Eq. (8)). Nguyen et al. (2021) generalized the vanilla sliced distances and max-sliced distances by introducing probability distributions on the hypersphere.

Lastly, it is well known that a larger number of samples leads to a better approximation of the one-dimensional Wasserstein distance, thus leading to a stronger generation performance (Nguyen et al., 2021; Lezama et al., 2021; Chen et al., 2022). Therefore, simply increasing the batch size can enhance the performance. However, the number of samples utilized for the model training is typically constrained by the memory required to store the model (Lezama et al., 2021). To overcome this limitation, Lezama et al. (2021) proposed an algorithm that evaluates the one-dimensional Wasserstein distance with a virtual large batch size, although they did not apply the algorithm to the training of generative models from scratch but to the fine-tuning of GAN models due to its high computational cost. This kind of algorithm may be useful even for the fine-tuning of SAN models especially when the obtained $h$ does not satisfy *separability* rigorously.

### E.3 LIMITATIONS

**Metrizable conditions.** Although our framework provides insights into GAN training without any unrealistic assumptions, a theoretical investigation into the *separability* is left as future work. In the current work, we conducted empirical studies on *separability* rather than a mathematical analysis. While our empirical observations on this property are consistent across different datasets, we believe additional theoretical investigation on it will be helpful for further understanding and improving GAN training.

**SAN.** One limitation of SAN is that it is not applicable to specific GANs (e.g., least-squares GAN) in a straightforward way. Naive least-squares SAN does not satisfy the metrizable condition, as its minimization problem does not satisfy the monotonic property regarding $R$ in Theorem 5.3. However, we should emphasize that SAN still covers a broad class of GAN objective including $f$-GANs, and it is also possible to construct a new SAN objective by adopting different GAN variants for $\mathcal{J}_{\mathrm{GAN}}$ and $\mathcal{V}_{\mathrm{GAN}}$.

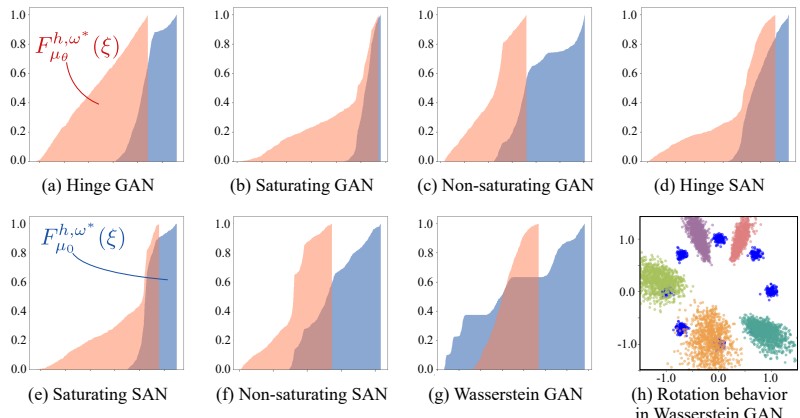

Figure 7: (a)–(g) Estimated cumulative density function at 5,000 iterations. *Separability* is satisfied in most GANs and SANs, but in the Wasserstein GAN, the discriminator does not satisfy *separability*. (h) Data samples and generated samples during training with Wasserstein GAN. Different colors represent different iterations (2,000, 4,000,..., 10,000). Rotational behavior is observed.

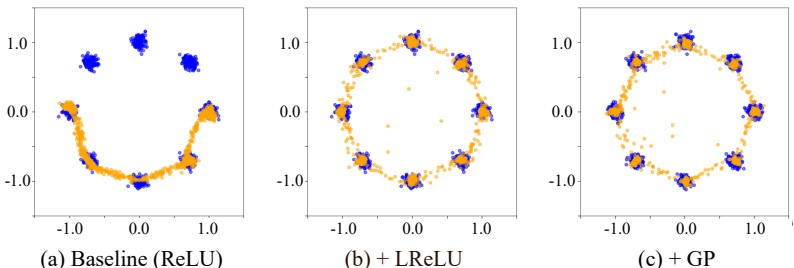

Figure 8: Effects of techniques to induce injectivity. The use of LReLU instead of ReLU and the addition of a GP to the maximization problem both improve the performance.

## F EXPERIMENTAL SETUP

### F.1 MIXTURE OF GAUSSIANS

In Sec. 7.1 and Appx. H, we use the same experimental setup as below. For both the generator and discriminator, we adopt simple architectures comprising fully connected layers by following the previous works (Mescheder et al., 2017; Nagarajan & Kolter, 2017; Sinha et al., 2020). We basically use leaky ReLU (LReLU) with a negative slope of 0.1 for the discriminator.

### F.2 IMAGE GENERATION

**DCGAN.** We borrow most parts of the implementation from a public repository[2]. Please refer to Tables 7 and 8 for details of the architectures. Spectral normalization is applied to the discriminator. We use the Adam optimizer (Kingma & Ba, 2015) with betas of $(0.0, 0.9)$ and an initial learning rate of 0.0002 for both the generator and discriminator. To convert GANs into SANs, we add two small modifications to the implementations. First, we apply a weight normalization instead of spectral normalization on the last linear layer of the discriminator, which maintains its exact weight on a hypersphere throughout the training. Next, we utilize the proposed objective functions for the maximization problems. We train the models for 2,000 iterations with the minibatch size of 64. We run the training with three different seeds for all the models and report the mean and deviation values in Table 3.

---

[2]https://github.com/christiancosgrove/pytorch-spectral-normalization-gan

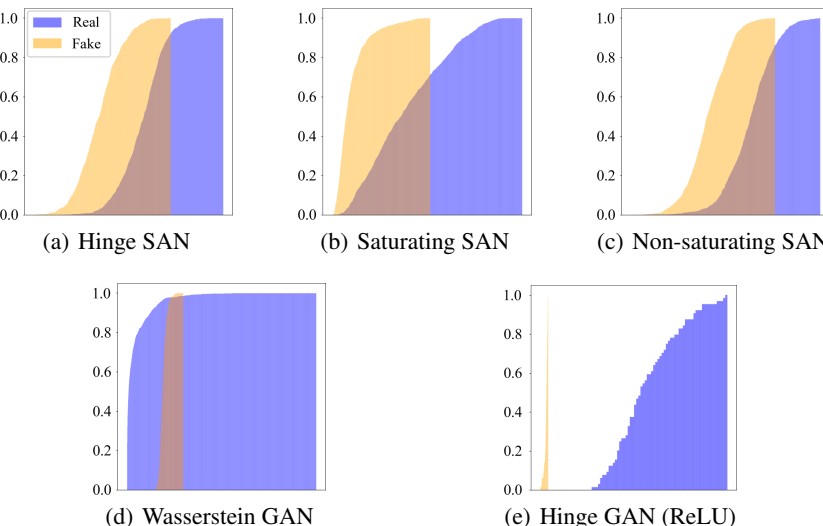

Figure 9: Estimated cumulative density functions for the MNIST case. We follow the same procedure as in Fig. 5 to plot them. (a)–(c) *Separability* is satisfied by the SAN models. (d) In contrast, Wasserstein GAN does not satisfy the property, leading to bad generation results (same as in Fig. 7(g)). (e) The use of ReLU in Hinge SAN does not destroy *separability* but leads to bad generation results due to the non-*injectivity* caused by the non-*injective* activation (see Fig. 10(a)).

Table 7: DCGAN architecture on CIFAR10.

| Generator | Discriminator |
|---|---|
| | Input: $x \in \mathbb{R}^{3 \times 32 \times 32}$ |
| Input: $z \in \mathbb{R}^{128} \sim \mathcal{N}(0, I)$ | 3×3, stride= 1, channel=64, LReLU |
| | SNConv (4×4, stride= 2, channel=64), LReLU |
| Deconv (4×4, stride=1, channel=512), BN, ReLU | SNConv (3×3, stride=1, channel=128), LReLU |
| Deconv (4×4, stride=2, channel=256), BN, ReLU | SNConv (4×4, stride=2, channel=128), LReLU |
| Deconv (4×4, stride=2, channel=128), BN, ReLU | SNConv (3×3, stride=1, channel=256), LReLU |
| Deconv (4×4, stride=2, channel=64), BN, ReLU | SNConv (4×4, stride=2, channel=256), LReLU |
| Deconv (4×4, stride=1, channel=3), Tanh | SNConv (3×3, stride=1, channel=512), LReLU |
| | SNLinear |

**BigGAN.** We use the authors' PyTorch implementation[3] to compare GANs and SANs in the setting of BigGAN on CIFAR10. We exactly follow the instructions in the repository to train the GAN in the class conditional generation task. The discriminator is constructed with ReLUs, which may impede *injectivity* as discussed in Appx. H.1. Since the aim of the experiment is to validate our training scheme presented in Sec 6 in existing settings, we do not change the activation function. We defer an ablation study on injectivity induction by the activation functions to Appx. H.3. Regarding the unconditional generation task, we simply replace the class embedding for the last linear layer with a single linear layer. Similarly to the DCGAN setting, we implement the SANs only by modifying the last linear layer of the discriminator and the maximization objective function. We run the training with three different seeds for respective cases and report the mean and deviation values in Table 4. Note that it has been reported that there are numerical gaps between the evaluation scores reported in the original paper and those obtained by reproduction with the PyTorch implementation (Song & Ermon, 2020).

**StyleGAN-XL.** We use the official implementation[4] to compare a GAN and a SAN in the setting of StyleGAN-XL. We follow the instructions in the repository to train StyleSAN-XL in the class

---

[3] https://github.com/ajbrock/BigGAN-PyTorch
[4] https://github.com/autonomousvision/stylegan-xl

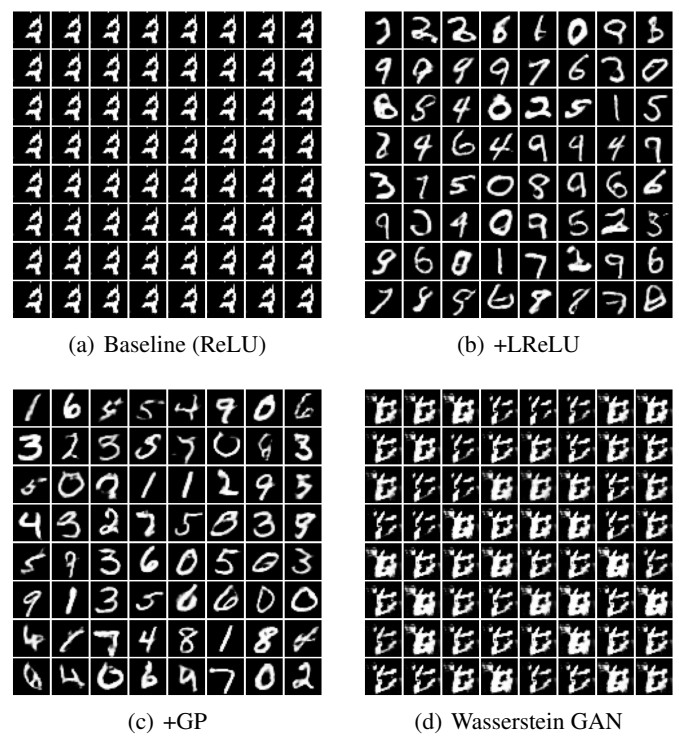

(a) Baseline (ReLU)  (b) +LReLU

(c) +GP  (d) Wasserstein GAN

Figure 10: (a)–(c) Effects of techniques to induce injectivity. (a) The use of ReLU leads to a collapse issue. (b) The use of LReLU instead of ReLU and (c) the addition of a GP to the maximization problem both improve the performance. (d) Generated results for Wasserstein GAN as a reference.

Table 8: DCGAN architecture on CelebA.

| Generator | Discriminator |
|---|---|
| | Input: $x \in \mathbb{R}^{3 \times 128 \times 28}$ |
| Input: $z \in \mathbb{R}^{128} \sim \mathcal{N}(0, I)$ | 3×3, stride= 1, channel=128, LReLU |
| Deconv (4×4, stride=1, channel=512), BN, ReLU | SNConv (4 × 4, stride= 2, channel=128), LReLU |
| Deconv (4×4, stride=2, channel=256), BN, ReLU | SNConv (3×3, stride=1, channel=256), LReLU |
| Deconv (4×4, stride=2, channel=128), BN, ReLU | SNConv (4×4, stride=2, channel=256), LReLU |
| Deconv (4×4, stride=2, channel=64), BN, ReLU | SNConv (3×3, stride=1, channel=512), LReLU |
| Deconv (4×4, stride=2, channel=32), BN, ReLU | SNConv (4×4, stride=2, channel=512), LReLU |
| Deconv (4×4, stride=2, channel=3), Tanh | SNConv (3×3, stride=1, channel=1024), LReLU |
| | SNLinear |

conditional generation task. As in the DCGAN and BigGAN settings, we implement the SAN by modifying the last linear layer of the discriminator and the maximization objective function. Besides, since just normalizing the last linear layer may deteriorate the representation ability of the overall network, we introduce the trainable scaling factor $s \in \mathbb{R}$ to keep the ability. Note that the scaling scalar is included in $h$ rather than $\omega$ to keep $\omega$ on the hypersphere, which is consistent with Listing 2. For ImageNet, we train a $256 \times 256$ generative model on top of the pretrained $128 \times 128$ generative model[5] in a progressive growing manner (Karras et al., 2018; Sauer et al., 2022). We set the `--kimg` option to 11,000 according to an author's comment in the repository[6]. We then report the FID and IS calculated with the provided code. For CIFAR10, we train a model from $16 \times 16$ resolution to the target resolution in a progressive way. We train a model with our arguments that

---

[5] https://s3.eu-central-1.amazonaws.com/avg-projects/stylegan_xl/models/imagenet128.pkl

[6] https://github.com/autonomousvision/stylegan-xl/issues/43#issuecomment-1111028101

Table 9: Numerical results for BigGAN (Brock et al., 2019) and BigSAN on CIFAR10 in unconditional setting.

| Method | FID ($\downarrow$) | IS ($\uparrow$) |
|--------|--------|--------|
| BigGAN | 17.16$\pm$1.34 | 8.42$\pm$0.11 |
| BigSAN | **14.45**$\pm$0.58 | **8.81**$\pm$0.04 |

Table 10: Numerical results for BigGAN (Brock et al., 2019) and the one fine-tuned by SAN training scheme.

| Method | FID ($\downarrow$) | IS ($\uparrow$) |
|--------|--------|--------|
| BigGAN (baseline) | 8.25$\pm$0.82 | 9.05$\pm$0.05 |
| BigGAN (fine-tuned with SAN) | **7.59**$\pm$0.23 | **9.04**$\pm$7.59 |

should be different from the original (see our repository [7] for the details of our training), resulting in a different model size.

# G  SUPPLEMENTARY EXPERIMENTS

## G.1  BIGGAN

In this section, we present two supplementary experimental results to further support the effectiveness of SAN training scheme.

**Unconditional BigSAN.**  We also compare SAN and GAN with BigGAN architecture in an unconditional generation case by following the experimental setting in Song & Ermon (2020). As reported in Table 9, SAN consistently improves the generation performance in terms of FID and IS.

**Fine-tuning GAN with SAN.**  Another interesting question is whether SAN training scheme is valid even for fine-tuning GAN. We fine-tuned the BigGAN models trained on CIFAR-10 with the SAN training objectives for 10k iterations. We found our proposed method is valid for fine-tuning existing GANs as reported in Table 10. Still, training the SAN from scratch benefits from our modification scheme more, leading to better performance than the fine-tuned model.

## G.2  STYLEGAN-XL ON FFHQ

Here, we train a model on FFHQ (Karras et al., 2019) from $16 \times 16$ resolution to $1024 \times 1024$ resolution in a progressive way. We train a model with our arguments that should be different from the original, resulting in a different model size (see our repository [9] for the details of our training). In this experiment, we calculate not only the FID but also the $FID_{CLIP}$[10], which is calculated in the feature space of a CLIP image encoder (Radford et al., 2021), because it has been reported that $FID_{CLIP}$ is more highly correlated with human assessment on FFHQ (Kynkäänniemi et al., 2023).

We report the numerical results in Table 11 and show generated images in Fig. 11. As shown in Table 11, StyleSAN-XL achieves comparable or better FID and $FID_{CLIP}$ scores. However, both StyleGAN-XL and StyleSAN-XL occasionally generate low-quality samples with poor proportions and artifacts, as observed in the original Projected GAN (Sauer et al., 2021). Kynkäänniemi et al. (2023) reported that StyleGAN2 (Karras et al., 2020) trained on FFHQ can produce high-quality and diverse samples, even though its FID score is not as good as that of StyleGAN-XL or StyleSAN-XL. We hypothesize that this discrepancy arises because the two encoders used in StyleGAN-XL/StyleSAN-XL training, EfficientNet (Tan & Le, 2019) and DeiT-base (Touvron et al., 2021), are trained on ImageNet, the same dataset used to train the Inception-V3 (Szegedy et al., 2016) utilized in FID calculation. While this may improve FID scores, it does not consistently enhance actual sample quality, particularly outside the dataset's domain.

---

[7]`https://github.com/sony/san`
[8]`https://s3.eu-central-1.amazonaws.com/avg-projects/stylegan_xl/models/ffhq256.pkl`
[9]`https://github.com/sony/san`
[10]`https://github.com/GaParmar/clean-fid`

Table 11: Numerical results for StyleGAN-XL (Sauer et al., 2022) and StyleSAN-XL on FFHQ. Scores marked with ‡ are reported in their repository. Note that our StyleSAN-XL models are larger in model size than StyleGAN-XL.

| Resolution | FID ($\downarrow$) | | FID$_{\text{CLIP}}$ ($\downarrow$) | |
| --- | --- | --- | --- | --- |
| | StyleGAN-XL[‡] | StyleSAN-XL[*] | StyleGAN-XL | StyleSAN-XL[*] |
| $256 \times 256$ | (2.19) | **1.68** | (**3.00**) | 3.01 |
| $512 \times 512$ | (2.23) | **1.77** | (3.21) | **3.06** |
| $1024 \times 1024$ | (2.02) | **1.61** | (3.62) | **3.11** |

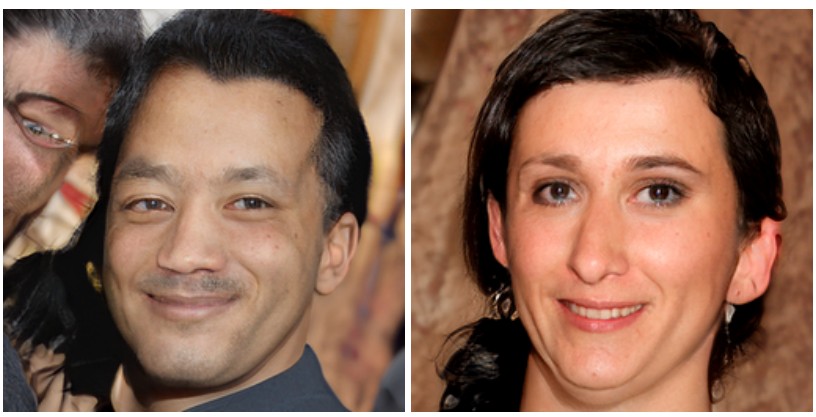

(a) StyleGAN-XL

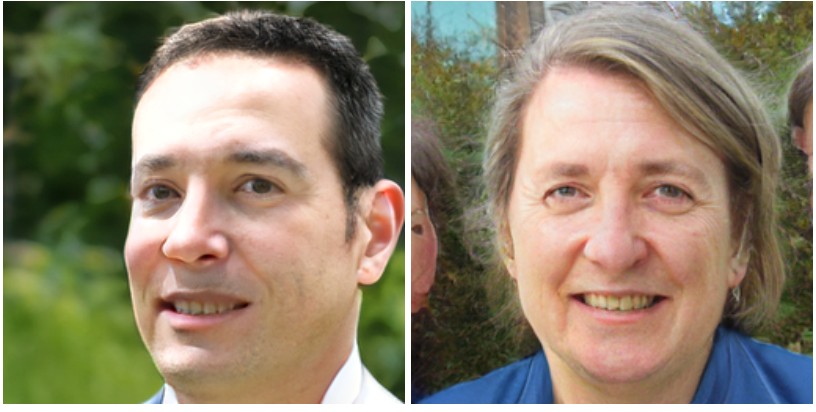

(b) StyleSAN-XL

Figure 11: Generated samples from StyleGAN-XL[8]and StyleSAN-XL trained on FFHQ.

## H  INVESTIGATION OF SEPARABILITY AND INJECTIVITY

In this section, *separability* and *injectivity* are empirically investigated.

### H.1  INDUCTIONS FOR INJECTIVITY

We present some potential inductions for injectivity, which are empirically verified in the next subsections.

There are various ways to impose *injectivity* on the discriminator, one of which is to implement the discriminator with an invertible neural network. Although this topic has been actively studied (Behrmann et al., 2019; Karami et al., 2019; Song et al., 2019), such networks have higher computational costs for training (Chen et al., 2022). Another way is to add regularization terms to the maximization problem. For example, a gradient penalty (GP) (Gulrajani et al., 2017) can

Table 12: Effect of injectivity induction by activation functions on FID and IS in the BigSAN setting. The use of LReLU leads to better or competitive scores than that of

| Metric | CIFAR10 | | CIFAR100 | |
|---|---|---|---|---|
| | BigSAN w/ ReLU | BigSAN w/ LReLU | BigSAN w/ ReLU | BigSAN w/ LReLU |
| FID ($\downarrow$) | $6.20\pm0.27$ | $5.97\pm0.12$ | $8.05\pm0.04$ | $7.97\pm0.06$ |
| IS ($\uparrow$) | $9.16\pm0.08$ | $9.23\pm0.03$ | $10.72\pm0.05$ | $10.84\pm0.03$ |

promote injectivity by explicitly regularizing the discriminator's gradient. The penalty enforces the norm of the discriminator Jacobian to be unity. This mitigates the occurrence of zero-gradient, one of the prominent factors contributing to non-injectivity.

In contrast, simple removal of operators that can destroy *injectivity*, such as ReLU activation, can implicitly overcome this issue. It is obvious that invertible activations (such as leaky ReLU) contribute to layerwise injectivity. Several studies suggest that constructing an injective neural network with ReLU is more challenging compared to invertible activation functions (Puthawala et al., 2022; Chan et al., 2023).

## H.2 MIXTURE OF GAUSSIANS

We visualize the generated samples mainly to confirm that the generator measures cover all the modes of the eight Gaussians. In addition, we plot the cumulative density functions of $\mathcal{S}^h I_{\mu_0}(\cdot, \omega)$ and $\mathcal{S}^h I_{\mu_\theta}(\cdot, \omega)$ to verify *separability*. As shown in Fig. 7, the cumulative density function for the generator measures trained with Wasserstein GAN does not satisfy *separability*, whereas the generator measures trained with other GAN losses do satisfy it. This may be what causes the rotation behavior of the Wasserstein GAN, as seen in Fig. 7(h).

To investigate the effect of *injectivity* on the training results, we train Hinge GAN using a discriminator with ReLU as a baseline. We apply two techniques to induce *injectivity*: (1) the use of LReLU and (2) the addition of a GP to the discriminator's maximization problem. As shown in Fig. 8, with either of these techniques, the training is improved and mode collapse does not occur.

## H.3 VISION DATASETS

We extend the experiment in Appx. H.2 to MNIST (LeCun et al., 1998) to investigate *injectivity* and *separability* on vision datasets. First, we train SANs with various objective functions and Wasserstein GAN on MNIST. We adopt DCGAN architectures and apply spectral normalization to the discriminator. As in Appx. H.2, we plot the cumulative density function for the generator measures trained with various objective functions in Fig. 9. The trained generator measures empirically satisfy *separability* except for the case of Wasserstein GAN, which is a similar tendency to the case of MoG. We also conducted an ablation study regarding the technique inducing *injectivity* by following Appx. H.2. We train Hinge SAN using a discriminator with ReLU as a baseline, and compare this baseline with two cases: (1) the use of LReLU and (2) the addition of a GP to the discriminator's objective function. The generated samples for these cases are shown in Fig. 10(a)–(c). As we can see, the generator trained with the ReLU-based discriminator suffers from the mode collapse issue, while in contrast, the techniques inducing *injectivity* mitigate the issue and lead to reasonable generation results. As a reference, we depict the generation result from a generator that is trained with a LReLU-based discriminator and Wasserstein GAN in Fig. 10(d), where we can see that the generated samples are completely corrupted. Note that the discriminator with ReLU satisfies *separability*, the same as in Fig. 9(e). This suggests that the deterioration in Hinge SAN with the ReLU-based discriminator (Fig. 10(a)) and that in the Wasserstein GAN model (Fig. 10(d)) are triggered by different causes, i.e., from non-injectivity and non-separability, respectively.

To further verify our insights in more realistic setting, we conduct another experiment using Big-GAN architecture, where the discriminator is constructed with ReLUs. For ablation of the injectivity induction by the activation functions, we replace ReLUs with LReLUs in the discriminator, and subsequently train BigSAN on both CIFAR-10 and CIFAR-100. The use of the injective activation leads to slightly better scores than the original BigSAN as reported in Table 12. This result is consistent with our theoretical and empirical investigations.

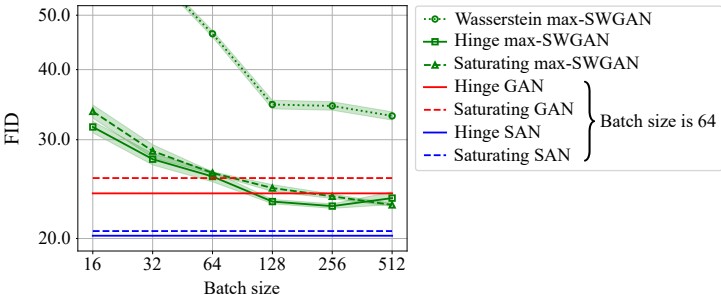

Figure 12: Empirical studies on the impact of batch size on SWGAN trained on CIFAR10. The FID score decreases as the batch size increases especially with $\mathcal{V}_{\mathrm{W}}$.

## I  INVESTIGATION OF $\mathcal{V}_{\mathrm{GAN}}$ IN SW

The aim of this empirical study is to compare $\mathcal{V}_{\mathrm{GAN}}$ in the SW context. To this end, we train max-SW-based generative models on CIFAR10 using the same experimental setting as in Sec. 7.2 (DCGAN). We apply the ASW to max-SWGAN (Deshpande et al., 2019), which was shown to achieve competitive or better generation performances compared to the uniform sampling of $\omega$ despite reducing the computational complexity and the memory footprint of the model. We train the model by the following optimization problem:

$$\max_{\omega,h} \mathcal{V}_{\mathrm{GAN}}(\langle \omega, h(x) \rangle), \quad \text{and} \tag{51a}$$

$$\min_{\theta} W_2(\mathcal{S}^h I_{\mu_0}(\cdot, \omega), \mathcal{S}^h I_{\mu_\theta}(\cdot, \omega)), \tag{51b}$$

Note that the maximizer $\omega$ in Eq. (51a) does not generally maximize the Wasserstein distance in Eq. (51b)[11]. Since the minimization problem (51b) requires sample sorting according to discriminator values to approximate the Wasserstein distance, the larger batch size is expected to enable a better distance approximation, leading to a better generation performance, at the expense of computational cost (refer to Fig. 6 in Nguyen et al. (2021) and Fig. 14 in Chen et al. (2022)). Regarding $\mathcal{V}_{\mathrm{GAN}}$, we utilize the maximization problems for Wasserstein GAN, hinge GAN, and saturating GAN (non-saturating GAN) for comparison.

We plot the FID score w.r.t. batch size in Fig. 12. Interestingly, with the maximization problem of Wasserstein GAN, the batch size greatly influences the generation performance, unlike the other GAN variants. We suspect the smaller sample size causes noisy approximation of the Wasserstein distance, since $\mathcal{V}_{\mathrm{W}}$ tends to bring non-separable $h$ (as shown in our experiments in Sec. H). In contrast, the other GAN variants lead to reasonable generative models even with a smaller batch size. The FID scores of GAN and SAN reported in Table 3 are also plotted here for reference. When the batch size is larger, the FID score of the max-SWGANs converges to a certain value, which is better than that of GAN but worse than that of SAN. We suspect this is because, among the comparison methods, SAN training is the only way to ensure *optimal direction*.

## J  GENERATED SAMPLES

Images generated by GANs and SANs are shown in Figs. 13, 14, 15, and 17.

---

[11]Although the idea of max-SW is to select the *optimal direction* (Deshpande et al., 2019), the proposed model does not satisfy the property due to its surrogate maximization objective $\mathcal{V}_{\mathrm{GAN}}$.

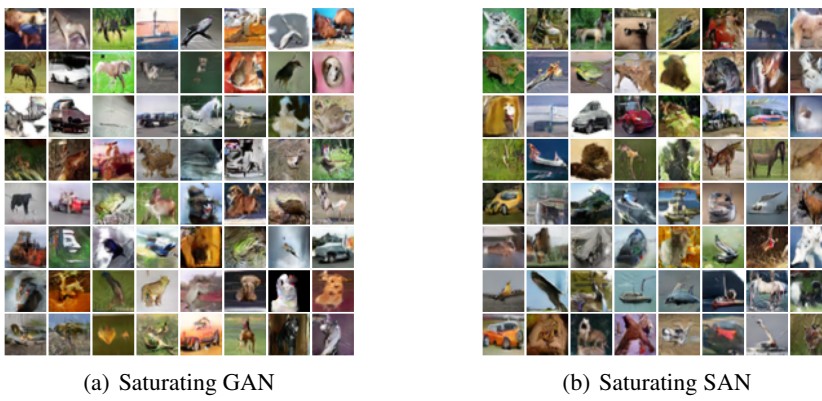

(a) Saturating GAN          (b) Saturating SAN

Figure 13: Generated samples from DCGAN trained on CIFAR10.

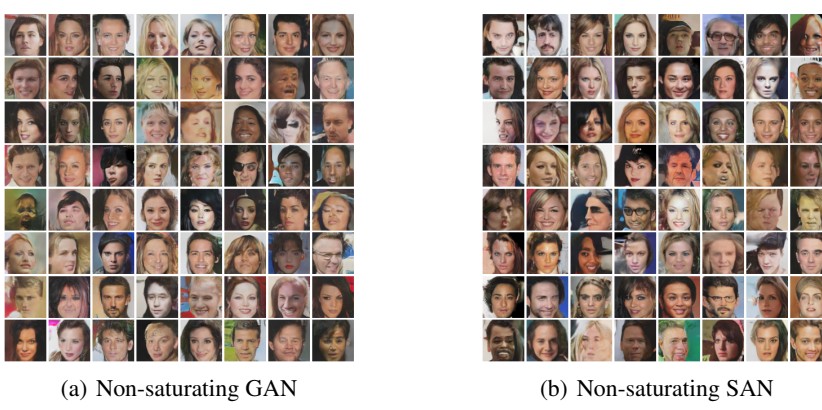

(a) Non-saturating GAN        (b) Non-saturating SAN

Figure 14: Generated samples from DCGAN trained on CelebA (128×128).

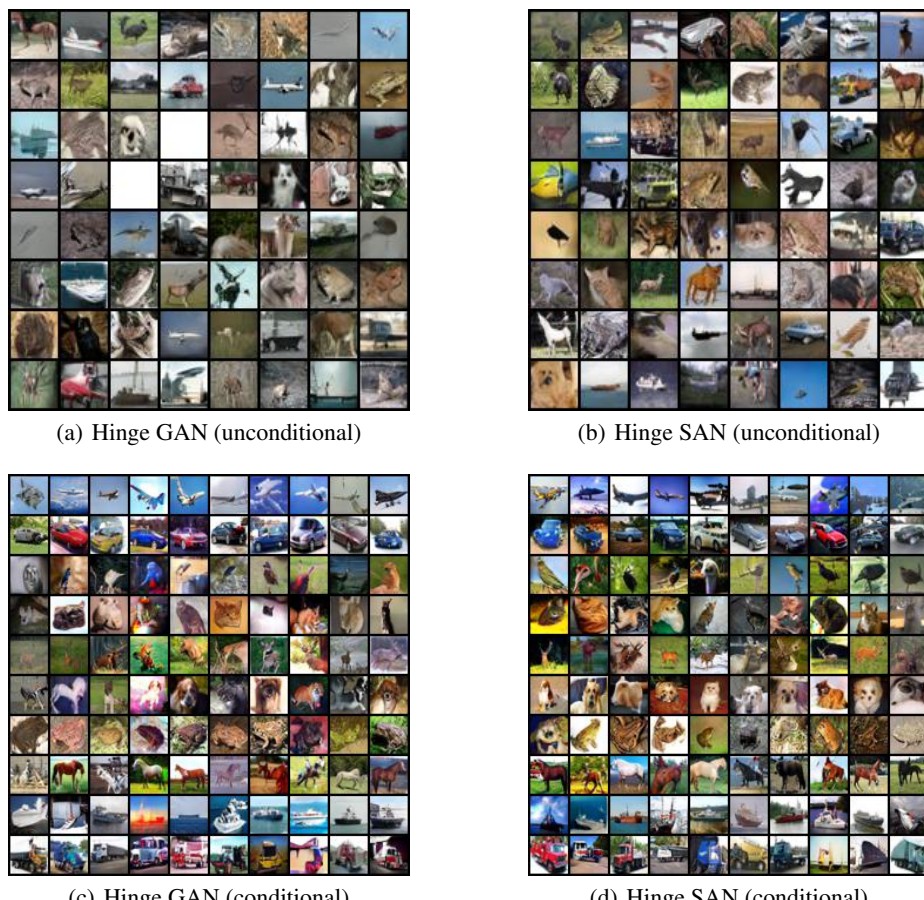

(a) Hinge GAN (unconditional)    (b) Hinge SAN (unconditional)

(c) Hinge GAN (conditional)    (d) Hinge SAN (conditional)

Figure 15: Generated samples from BigGAN trained on CIFAR10.

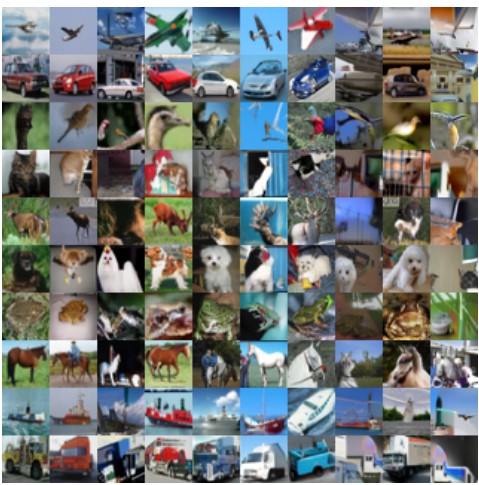

Figure 16: Generated samples from StyleSAN-XL trained on CIFAR10.

---

[12]`https://s3.eu-central-1.amazonaws.com/avg-projects/stylegan_xl/models/imagenet256.pkl`

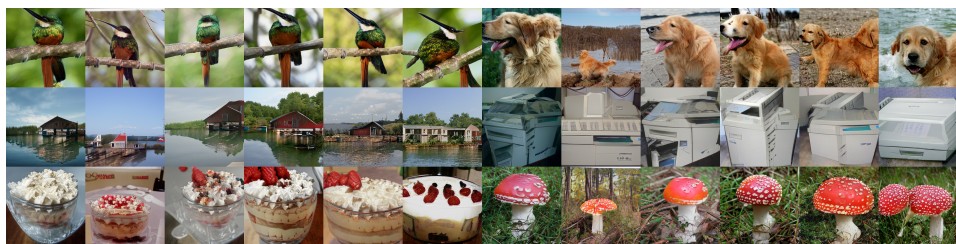

(a) StyleGAN-XL

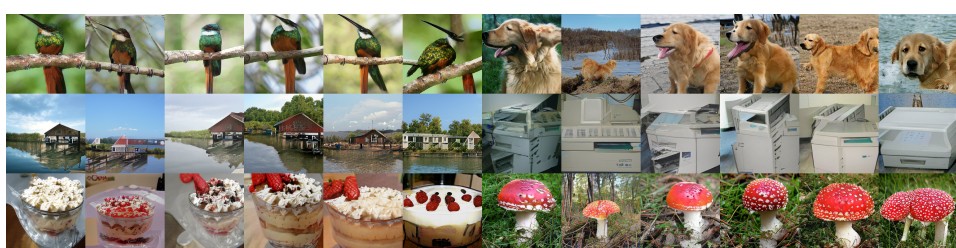

(b) StyleSAN-XL

Figure 17: Generated samples from StyleGAN-XL[12]and StyleSAN-XL trained on ImageNet.

