# OpenReview forum: "SAN: Inducing Metrizability of GAN with Discriminative Normalized Linear Layer"
_ICLR.cc/2024/Conference — ICLR 2024 poster_

### Official Review · Reviewer_EquL · 2023-10-25

**Soundness:** 3 good
**Presentation:** 3 good
**Contribution:** 2 fair
**Rating:** 6
**Confidence:** 4

**Summary:**

This article introduces a novel theoretical perspective to shed light on the optimization process, which determines whether the generator receives gradients that bring its distribution closer to the target distribution. The article introduces three metrizable conditions: direction optimality (evaluating the gradient between two distributions), separability (examining the distinctness between two distributions), and injectivity (measuring the metric as a distance). These conditions aim to provide clarity on the underlying motivation. Furthermore, the article includes a theoretical proof and introduces a new loss function and a normalized linear layer. These enhancements are designed to boost the discriminative capability of the discriminator and enable the generation of diverse, high-fidelity samples from the generator.

**Strengths:**

This article introduces a novel theoretical perspective aimed at addressing whether the optimization process truly provides the generator with gradients that lead its distribution closer to the target distribution. The article proposes three metrizable conditions, namely direction optimality (evaluating the gradient between two distributions), separability (evaluating disjointedness), and injectivity (measuring the metric as a distance).

Additionally, the article provides both theoretical and empirical evidence to demonstrate that only Wasserstein GANs fulfill these conditions. To enhance the discriminative capability of the discriminator and facilitate the generation of diverse, high-fidelity samples from the generator, the authors introduce a new loss function and a normalized linear layer. This normalized linear layer serves as a plug-in component that can be seamlessly incorporated into various GAN models.

In experimental evaluations, the SAN model achieves the best FID score when compared to numerous GAN baseline models.

**Weaknesses:**

The article's motivation, which revolves around the examination of whether trained discriminators effectively supply gradient information to optimize generators for reducing dissimilarities, appears to have been explored in prior articles [1-5]. However, the article falls short in providing a clear connection between the theoretical results and their practical implications. Moreover, the significance of the theoretical findings might benefit from more detailed explanation.


1.Variational inference via Wasserstein gradient flows

2.Variational Wasserstein gradient flow

3.Deep Generative Learning via Variational Gradient Flow

4.A Framework of Composite Functional Gradient
Methods for Generative Adversarial Models

5.Gradient Layer: Enhancing the Convergence of Adversarial Training for Generative Models

**Questions:**

I have read the paper carefully, so i want the author to clarify the questions as follows:

If the author can clarify the following questions well, I'd like to raise my rating score.

1. If I understand correctly, "direction optimality" implies that a negative gradient of the metric between the generated distribution and the target distribution , which guides the generated distribution toward the target. "Separability" pertains to two disjointed distributions, and "injectivity" suggests that the metric between the generated and target distributions is a distance. Is this interpretation accurate?

2. Could you clarify the symbol "$*$" in Table 1?


3. I recommend that when the article first mentions metrizable conditions, namely, direction optimality, separability, and injectivity, it would be more reader-friendly to provide an intuitive understanding of these terms and indicate which paragraphs explain these definitions. This would help readers quickly grasp the key concepts before delving into the detailed explanations.

4. I have a question about the statement as follows:
most existing GANs besides Wasserstein GAN do not satisfy direction optimality with the maximizer $w$ of $\mathcal{V}$. Although most existing GANs do not  satisfy direction optimality proposed in the article, but the discriminator of these gans actually provides the generator with gradients that make its distribution close to the target distribution which has been discussed in other articles[1-5]. What do you think about this question?



5. There might be an error in the symbol  $g\# $     $=\sigma(g^{-1}(B))$.

The function $g$ is defined as $g: Z \rightarrow X$, so the inverse function $g^{-1}$ should be $g^{-1}: X \rightarrow Z$. The term $\sigma(\cdot) \in \mathcal{P}(Z)$. However, $g\#$

is meant to represent the generator $g: Z \rightarrow X$. Thus, there appears to be a discrepancy.


6. I recommend that the author provide a clear explanation of why Equation (16) satisfies Theorem 5.3. I find it somewhat confusing and would appreciate a more comprehensible clarification.

7. I have reviewed the code in the supplementary materials. There is a discrepancy between the code provided in Listing 2 (SAN discriminator) and lines 92-96 in the file "STYLESAN-XL/pg_modules/san_mondules.py/SANConv2d." To ensure accuracy, please verify which version is the correct one.
8. Should the SAN be trained from scratch, or can it be fine-tuned, especially in the last layer, using the Eq.16 loss function?

9. I find it quite confusing to understand the importance of the three conditions in your theorem 5.3. In my opinion, direction optimality should be the most important condition, while the separability and injectivity conditions may not carry the same weight. This is because metrics used in GANs, such as f-divergences or IPM functions, inherently satisfy the distance property. Maintaining the separability condition becomes increasingly challenging as GAN training progresses and the generated distribution approaches the real data distribution.
Therefore, I recommend that the author clearly outline the relationships and significance of all three conditions to provide a better understanding.


1.Variational inference via Wasserstein gradient flows

2.Variational Wasserstein gradient flow

3.Deep Generative Learning via Variational Gradient Flow

4.A Framework of Composite Functional Gradient
Methods for Generative Adversarial Models

5.Gradient Layer: Enhancing the Convergence of Adversarial Training for Generative Models

---

> ### Author Response · Authors · 2023-11-17
> **Author reply (1/5)**
>
> Thank you for reviewing our paper carefully. Your comments and questions are valuable to improve our work. We have carefully considered each of your points and have provided our response below.
>
> **Weakness**
> ---
> Thank you for suggesting the papers. We have carefully checked the papers, and found some of them are helpful to highlight the uniqueness of our theoretical perspective more.
>
> We would like to first clarify the problem we are tackling to make it clear the difference between our work and the suggested papers. Since practical GAN training cannot exactly maximize the inner problem $\mathcal{V}$ w.r.t. the discriminator, we aim to provide a sufficient condition for **discriminators to function as certain distances**, which is more feasible during the training. The new theoretical framework relieves us from an impractical assumption that discriminators keep optimal throughout the training.
>
> [5] is a pioneering work that first introduces the concept of gradient flows in the GAN problem. The authors suspected that one reason of difficulty of GAN training comes from limitation of the representational power of the generator. Hence they proposed to insert gradient layers into the generator to enhance the representational power. They conducted convergence analysis based on smoothness analysis of generator loss, which does not assume optimal discriminator. However, the theoretical results **do not care about the property of the convergent points**. The work did not discuss whether trained discriminators with the proposed method indeed provide generator optimization with gradients that reduce dissimilarities. [4] is another work improving GAN training with the concept of functional gradient learning. They explicitly assume that **discriminators are optimal**.
>
> [2-3] proposed training schemes of generative models based on variational gradient flow, which is different from but related to GAN. They rely on the primal form of the original divergence minimization problem unlike usual GANs. They still use the discriminator as an **ideal** density estimator, which is obtained by adversarial training as in GANs. It means the proposed algorithm is based on the assumption that **the discriminator keeps optimal**, which is required for having such an ideal density estimator.
>
> Lastly, [1] deals with the problem of variational inference, and is less relevant to the topic of GAN.
>
> We appreciate your listing the papers again. We have added the references [2-4] to the second paragraph of Appendix E.2.
>
> [1] Variational inference via Wasserstein gradient flows
>
> [2] Variational Wasserstein gradient flow
>
> [3] Deep Generative Learning via Variational Gradient Flow
>
> [4] A Framework of Composite Functional Gradient Methods for Generative Adversarial Models
>
> [5] Gradient Layer: Enhancing the Convergence of Adversarial Training for Generative Models

---

> ### Author Response · Authors · 2023-11-17
> **Author reply (2/5)**
>
> **Preliminary comments**
> ---
> As you encouraged throughout the feedback, providing concise explanations and intuitions regarding the metrizable conditions more should make our manuscript easier for wider readers to follow. Here, we would like to describe them, prior to addressing the individual questions.
>
> First, we briefly explain how each of the metrizable conditions works in our theorem by taking Wasserstein GAN as an example. We believe this is helpful to get why the three conditions are crucial. Our strategy to derive Theorem 5.3 is basically to connect GAN problem with a variant of sliced Wasserstein, max-ASW, by using the novel framework, a functional mean divergence\* (FM\*), as depicted in Figure 1. First, *direction optimality* ensures that $\mathcal{J}\_{Wass}$ corresponds to a minimization objective for FM\*. Next, *separability* guarantees the equivalence of FM${}^*$ to the max-ASW. Now, thanks to the two conditions, the max-ASW can be minimized by $\mathcal{J}\_{Wass}$. Lastly, *injectivity* is needed to ensure that the max-ASW is indeed a distance.
>
> Next, we briefly provide our intuitions to the conditions. With the representation of the discriminator $f(x)=\langle\omega,h(x)\rangle$, it is interpreted as extracting features $h(x)$ and projecting them onto a one-dimensional space with the direction $\omega$ (please also refer to the final paragraph of Section 6). *Injectivity* and *separability* are conditions for the feature extractor $h(x)$.
> 1. *Injectivity* literally indicates that of function $h(x)$, aimed at preventing the loss of information from the original real/fake samples.
> 2. *Separability* is characterized by the cumulative density functions of $\mathcal{S}^hI_{\mu_{\theta}}(\cdot,\omega)$ and $\mathcal{S}^hI_{\mu_0}(\cdot,\omega)$ . This condition ensures that optimal transport maps for all the samples from $\mathcal{S}^hI_{\mu_{\theta}}(\cdot,\omega)$ to $\mathcal{S}^hI_{\mu_0}(\cdot,\omega)$ share the same direction. We will elaborate an illustration regarding *separability*.
> 3. *Direction optimality* is a condition for $\omega$ to project the features with a normalized projection that most effectively distinguishes the sets of real and fake samples.
>
> We have incorporated the above explanations and additional illustrations into the revised manuscript in order to convey the key concept in a more intuitive way. For the details about how we have reflected the above on the manuscript, please refer to our comments on your questions below.

---

> > ### Comment · Reviewer_EquL · 2023-11-18
> > **Reviewer reply**
> >
> > Thank you for responding to my comment carefully.  I read the revision of the manuscript, and realized the  insight idea of your work. So I raised my review score in the system. I have a last question as follows: Why do you call the theorem, which ensure the distance of a max-ASW, "injectivity"? I can not get any relationship between the distance function and the name  "injectivity".

---

> > > ### Author Response · Authors · 2023-11-18
> > > **Author reply**
> > >
> > > Thank you very much for taking the time to review our paper and for considering our responses. We sincerely appreciate your efforts in reassessing the manuscript and adjusting the score.
> > >
> > > We would be happy to answer your additional question. The condition *injectivity* in our theorem indicates that $h$ should be an injective function. The injective $h(x)$ maps distinct elements in $X$ to distinct elements, i.e., $\forall x_1,x_2\in X$, $x_1\neq x_2\Longrightarrow h(x_1)\neq h(x_2)$. This property guarantees that max-ASW given by such $h$, $\textit{max-ASW}_h(\cdot,\cdot)$, is indeed a distance (Lemma 4.4). In summary, **injectivity is the condition regarding $h$ that ensures the distance of the max-ASW**.
> > >
> > > In response to the question, we have added modifications to Figure 3 so that readers can easily get that *injectivity* and *separability* are conditions for $h$ whereas *direction optimality* is the condition for $\omega$. Thank you for your further question again.
> > >
> > > We always appreciate additional comments and suggestions.

---

> ### Author Response · Authors · 2023-11-17
> **Author reply (3/5)**
>
> **Questions**
> ---
> > 1. If I understand correctly, "direction optimality" implies that a negative gradient of the metric between the generated distribution and the target distribution , which guides the generated distribution toward the target. "Separability" pertains to two disjointed distributions, and "injectivity" suggests that the metric between the generated and target distributions is a distance. Is this interpretation accurate?
>
> Thank you for elaborating your intuition. The interpretations regarding *direction optimality* and *injectivity* are correct. *Optimal direction* corresponds to the direction increasing the distance most. *Injectivity* ensures the metric induced by the discriminator is indeed a distance, as you mentioned.
>
> *Separability* and “disjointed distributions” are disconnected somewhat. Whether the distributions of $h(x)$ with $x\sim\mu_{\theta}$ and $x\sim\mu_0$ are disjoint depends more on whether $\mu_0$ and $\mu_\theta$ are disjoint from each other. *Separable* h(x) does not necessarily mean such disjoint distributions in the feature space.
>
> The comment inspired us to provide our intuitive explanations for the conditions. Please kindly refer to our comment in the thread **Preliminary comments**. We have put the intuitions in the revised manuscript as below.
>
> 1. *direction optimality* right after Proposition 3.5,
>
> 2. *separability* right after Lemma 4.3, and
>
> 3. *injectivity* right before Lemma 4.4
>
> Furthermore, we have incorporated additional figure (Figure 1 in the revised manuscript), which illustrates *direction optimality* and *separability*.
>
> ---
> > 2. Could you clarify the symbol "\*" in Table 1?
>
> Thank you for the question. We apologize for the lack of clarity about the symbol. Whether the trained discriminators are injective is **not directly affected by the selected maximization objective** but by other factors such as discriminator design (e.g., activation functions) and regularization objectives for discriminators. Because the aim of Table 1 is mainly to show the inductive effects of the GAN objectives on the metrizable conditions, we refrained from using cross and check marks in the injectivity column. We have clarified it in the caption of Table 1 of the updated manuscript.
>
> ---
> > 3. I recommend that when the article first mentions metrizable conditions, namely, direction optimality, separability, and injectivity, it would be more reader-friendly to provide an intuitive understanding of these terms and indicate which paragraphs explain these definitions. This would help readers quickly grasp the key concepts before delving into the detailed explanations.
>
> Thank you for the suggestion. We have reflected the feedback on the new manuscript. Please kindly refer to our reply to the first question for how we have reflected.
>
> ---
> > 4. I have a question about the statement as follows: most existing GANs besides Wasserstein GAN do not satisfy direction optimality with the maximizer $w$ of $\mathcal{V}$. Although most existing GANs do not satisfy direction optimality proposed in the article, but the discriminator of these gans actually provides the generator with gradients that make its distribution close to the target distribution which has been discussed in other articles[1-5]. What do you think about this question?
>
> Please refer to our reply to **weakness**. There is no guarantee that practical (suboptimal) discriminators, optimized by existing GANs, provide the generator with such desirable gradients. We formulate the fundamental problem of whether suboptimal discriminators can still yield such gradients, by introducing a novel concept: the metrizability of discriminators (see Definition 1.1). We address this question without imposing the optimal discriminator assumption.
>
> ---
> > 5. There might be an error in the symbol $g\sharp=\sigma(g^{-1}(B))$.
>
> Thank you for the detailed check. We use the current notation for pushforward of probability measure by following the textbook and previous papers listed below. We would follow your suggestion if our notation has errors actually or you could teach us better notation.
>
> Villani, “Optimal transport: old and new,” Springer, 2009
>
> Chu et al., “Smoothness and Stability in GANs,” ICLR, 2020
>
> Rahman et al., “Generative Adversarial Neural Operators,” TMLR, 2022
>
> Antoine et al., “Can Push-forward Generative Models Fit Multimodal Distributions?,” NeurIPS, 2022

---

> ### Author Response · Authors · 2023-11-17
> **Author reply (4/5)**
>
> ---
> > 6. I recommend that the author provide a clear explanation of why Equation (16) satisfies Theorem 5.3. I find it somewhat confusing and would appreciate a more comprehensible clarification.
>
> We acknowledge your suggestion, and believe it will make the idea behind our loss design clearer. As you suggested, we have put a sentence right after Equation (16) to explain that the first and second terms in Equation (16) respectively induce *separability* and *direction optimality*.
>
> More intuitively, the loss design can be explained by our observations summarized in Table 1: (1) we employ the general GAN maximization objective, distinct from Wasserstein GAN, to induce the *separability*, and (2) Wasserstein GAN’s maximization objective, with some modifications according to $\mathcal{J}_{GAN}$, is used for $\omega$. Please note that, as in our reply to the Question 2, *injectivity* is not directly affected by the maximization objective but induced by the discriminator architecture design.
>
>
> ---
> > 7. I have reviewed the code in the supplementary materials. There is a discrepancy between the code provided in Listing 2 (SAN discriminator) and lines 92-96 in the file "STYLESAN-XL/pg_modules/san_mondules.py/SANConv2d." To ensure accuracy, please verify which version is the correct one.
>
> We sincerely appreciate your checking our source code carefully. Our StyleSAN-XL implementation is consistent with the pseudo code in Listing 2, and there is no essential gap between them. We have added a supplementary explanation as described below to Appendix F.2.
>
> In the implementation, the discriminator is implemented as $f(x)=s\langle\omega,\tilde{h}(x)+b\rangle$, where $s$ is a trainable scalar and $\tilde{h}(x)+b$ is an input of the last convolutional layer. This discriminator is equivalent to $f(x)=\langle\omega,h(x)\rangle$ with $h(x)=s(\tilde{h}(x)+b)$ because of linearity of the inner product. Since just normalizing the last linear layer may deteriorate the representation ability of the overall network, we introduced the scaling factor $s$ to keep the ability. Even with the scaling factor, the discriminator implementation still follows the concept of SAN. We calculated the discriminator outputs with $\omega$ detached for $\langle\omega^-,h\rangle$, while detached both $s$ and $\tilde{h}(x)+b$ to get $\langle\omega,h^-\rangle$.
>
> ---
> > 8. Should the SAN be trained from scratch, or can it be fine-tuned, especially in the last layer, using the Eq.16 loss function?
>
> Thank you for bringing the interesting perspective. This should be another interesting topic. In response to this comment, we conducted an additional experiment to confirm the SAN training scheme is effective even for fine-tuning GAN. We fine-tuned the BigGAN models trained on CIFAR-10 with the SAN training objectives for 10k iterations. We found our proposed method is valid for fine-tuning existing GANs as reported below. Still, training the SAN from scratch benefits from our modification scheme more, leading to better performance than the fine-tuned model. We have put the result to Appendix G. in the revised manuscript to show the potential of SAN for fine-tuning GAN. Thanks again for your valuable comment.
>
> | CIFAR10 | FID ($\downarrow$) | IS ($\uparrow$) |
> | --- | --- | --- |
> | BigGAN (baseline) | 8.25+/-0.82 | 9.05+/-0.05 |
> | BigGAN (fine-tuned with SAN) | 7.59+/-0.23  | 9.04+/-0.08 |

---

> ### Author Response · Authors · 2023-11-17
> **Author reply (5/5)**
>
> ---
> > 9-1. I find it quite confusing to understand the importance of the three conditions in your theorem 5.3. In my opinion, direction optimality should be the most important condition, while the separability and injectivity conditions may not carry the same weight.
>
> Thank you for the comment. *Direction optimality* is the most important condition for our methodology part because the idea behind SAN is to impose the property on arbitrary GANs. We have stressed that *direction optimality* is particularly crucial for our methodological proposal, right after presenting Theorem 5.3 in the revised manuscript. Adding such a note near to the theorem should be helpful for readers to find the connection of Theorem 5.3 and the methodology section (Section 6). For explanations for how each condition works in the theorem, please refer to our comment in the thread **Preliminary comments**.
>
> ---
> > 9-2. This is because metrics used in GANs, such as f-divergences or IPM functions, inherently satisfy the distance property.
>
> Although GAN formulation is based on dual formulation of such metrics, the distances can only be evaluated using the **optimal discriminators**. While the optimal discriminator assumption is convenient for theoretical analysis of GANs, achieving such optimality in practical cases is rarely feasible. In summary, there is no guarantee that practical GANs satisfy the distance property. As explained in our comment on the **weakness**, the aim of our work is to answer the question whether the discriminator can serve as a distance without relying on such strong assumptions.
>
> ---
> > 9-3. Maintaining the separability condition becomes increasingly challenging as GAN training progresses and the generated distribution approaches the real data distribution.
>
> We agree that the *separability* condition becomes increasingly difficult to achieve as the generator distribution approaches the data distribution. However, as long as $\mu_{\theta}\neq\mu_0$, a class of *separable* functions $h$ includes a broad range of functions, whereas the optimal discriminator is always unique for the given measures $(\mu_{\theta},\mu_0)$. In this sense, the optimal discriminator is much harder than the metrizable conditions to achieve.
>
> ---
> > 9-4. Therefore, I recommend that the author clearly outline the relationships and significance of all three conditions to provide a better understanding.
>
> We appreciate your valuable suggestions. We have put an illustration (Figure 3 of the updated manuscript) in the updated manuscript to clearly show the relationship and significance. Lastly, we have reflected your suggestions regarding the three key conditions on the manuscript, but we would appreciate your further feedback if there still remains something unclear.

---

### Official Review · Reviewer_xjKk · 2023-11-01

**Soundness:** 3 good
**Presentation:** 2 fair
**Contribution:** 3 good
**Rating:** 6
**Confidence:** 4

**Summary:**

The authors focus on evaluating the metrizability of discriminators in Generative Adversarial Networks (GANs) training by determining if the discriminator's measure can constitute a metric, defined by properties such as directionality, separability, and injectivity. Initially, they establish a connection between the Wasserstein distance and the Functional Mean Divergence (FM*). Then the Separability is proposed to link FM* to the maximum Average Sliced Wasserstein (max-ASW) distance. Subsequently, they introduce the injectivity for the design of effective discriminators. They examine commonly used GAN frameworks, including Wasserstein GAN and other losses GANs, and find that most GAN architectures, including those using Hinge-loss, Saturating, and Non-Saturating loss functions, typically do not fulfill the criterion of directional optimality. In response to this, the authors propose a straightforward modification to enhance this property within these GANs. To substantiate their theoretical claims, they conduct experiments that demonstrate the significance of these three properties. They also apply their modifications to contemporary models such as DCGAN, BigGAN, and StyleGAN-XL. The results affirm that their simple yet effective alterations lead to improvements over the baseline performance, thereby confirming the practical value of their theoretical insights.

**Strengths:**

1. The authors present a theoretically grounded analysis that yields a straightforward method for enhancing GAN discriminators.

2. Their theoretical contributions are insightful and offer practical guidance for discriminator design.

3. The experiments conducted are well-aligned with the theory, and the clarity of the writing effectively conveys the study's findings and implications.

**Weaknesses:**

1. While the study shows experimental progress, the improvement in FID (Frechet Inception Distance) is modest.

2. The authors are encouraged to conduct experiments on CIFAR-100 using BigGAN and FFHQ using StyleGAN2 to further verify the effectiveness of the proposed method.

3. To enhance accessibility, the authors should consider simplifying the mathematical notation to cater to a wider audience.

**Questions:**

1. Could the authors cite studies or evidence that show how regularizing the gradient of the discriminator or switching from ReLU to LeakyReLU might enhance injectivity?

2.  In Table 4, are the results shown for BigGAN the ones obtained after replacing ReLU with LeakyReLU?

---

> ### Author Response · Authors · 2023-11-17
> **Author reply (1/2)**
>
> We greatly appreciate the valuable feedback you have provided, and have addressed each of your comments. Please kindly find our response below.
>
> **Weakness**
> ---
> > 1. While the study shows experimental progress, the improvement in FID (Frechet Inception Distance) is modest.
>
> We believe our empirical progress in terms of FID is nontrivial despite the fact that our proposed method does not increase computational complexity. Additionally, it is noteworthy that our simple method consistently yields improvements across all the experimental configurations.
>
> In the case of DCGAN and BigGAN, the enhancement in FID is substantial, exhibiting a significant improvement of more than 15% on average. In addition, our simple modification scheme enhances even the SOTA GAN, StyleGAN-XL, by around 8% in FID. This is particularly surprising given that StyleGAN-XL has already integrated a bunch of techniques aimed at stabilization and improvement. The result on ImageNet-256 shows the effectiveness of SAN even on a large-scale setup.
>
> ---
> > 2. The authors are encouraged to conduct experiments on CIFAR-100 using BigGAN and FFHQ using StyleGAN2 to further verify the effectiveness of the proposed method.
>
> We agree that expanding the evaluation using additional datasets could provide more comprehensive validation of the proposed method's effectiveness. To further extend our experiments, we additionally trained GAN and SAN on CIFAR-100 using BigGAN architecture. We followed the same hyperparameter setting (e.g., learning rate) as employed for CIFAR-10, yielding the numerical results reported below. SAN exhibits superior performance. Especially, FID has been significantly improved by 25%. We have added the results to Section 7.2 in the updated manuscript..
>
> | CIFAR100 | FID ($\downarrow$) | IS ($\uparrow$) |
> | --- | --- | --- |
> | BigGAN | 10.73+/-0.16 | 10.56+/-0.01 |
> | BigSAN | **8.05**+/-0.04  | **10.72**+/-0.05 |
>
> ---
> > 3. To enhance accessibility, the authors should consider simplifying the mathematical notation to cater to a wider audience.
>
> We agree that some notations are a bit complicated. We have simplified the following notations and their combinations in the revised manuscript. Besides, we have added Table 6 to summarize some key notations, which are not introduced in Section 2.1.
>
> - $\mathcal{J}\_{\text{Wass}}$ $\Longrightarrow$ $\mathcal{J}_{W}$
>
> - $\mathcal{D}^{\text{max-ASW}}\_{\mathcal{F}}$ $\Longrightarrow$ $\mathcal{D}^{\text{mA}}_{\mathcal{F}}$
>
> - $\mathcal{D}\_{\mathcal{F}\_{\text{Inj}}}$ $\Longrightarrow$ $\mathcal{D}\_{\mathcal{F}\_{\text{I}}}$
>
> - $\mathcal{D}\_{\mathcal{F}\_{\text{Sep}(\mu,\nu)}}$ $\Longrightarrow$ $\mathcal{D}\_{\mathcal{F}\_{\text{S}}}$
>
> - $\mathcal{D}\_{\mathcal{F}\_{\text{Inj}}\cap\mathcal{F}\_{\text{Sep}(\mu,\nu)}}$ $\Longrightarrow$ $\mathcal{D}\_{\mathcal{F}\_{\text{I,S}}}$

---

> ### Author Response · Authors · 2023-11-17
> **Author reply (2/2)**
>
> **Questions**
> ---
> > 1. Could the authors cite studies or evidence that show how regularizing the gradient of the discriminator or switching from ReLU to LeakyReLU might enhance injectivity?
>
> Thank you for the valuable feedback. We have added some explanations and references to Appendix H.1 in the updated manuscript as below.
>
> The gradient penalty serves as a regularization mechanism, enforcing the norm of the discriminator Jacobian to be unity. This mitigates the occurrence of zero-gradient, one of prominent factors contributing to non-injectivity. While no suitable citations for the gradient penalty are available, we would like to add such a brief explanation.
>
> Concerning activation functions, it is evident that invertible activations (such as leaky ReLU) contribute to layerwise injectivity. We will cite the following papers, which suggest constructing an injective neural network with ReLU is more challenging compared to invertible activation functions.
>
> [1] Puthawala et al., “Globally injective ReLU networks,” JMLR, 2022
>
> [2] Chan et al., “LU-Net: Invertible Neural Networks Based on Matrix Factorization,” 2023
>
> ---
> > 2. In Table 4, are the results shown for BigGAN the ones obtained after replacing ReLU with LeakyReLU?
>
> We deeply appreciate your keen observation. The discriminator of the original BigGAN is constructed with ReLUs. As our main goal of the experiment is to validate our proposed training scheme proposed in Section 6, we did not change any other conditions including the activation functions. We have clarified this in Appendix F.2 of the revised manuscript.
>
> Inspired by the question, we replaced ReLUs with leakyReLUs in the discriminator, and subsequently trained BigSAN on both CIFAR-10 and CIFAR-100, for ablation of the injectivity induction by the activation functions. We got the results listed in the table below, showing that the use of leakyReLU leads to better or competitive scores than that of ReLU in both cases. The empirical result is also insightful as it is consistent with our claim. We have put the experimental results in Appendix H.3 of the revised manuscript. Thanks again for your valuable comment.
>
> | CIFAR10 | FID ($\downarrow$) | IS ($\uparrow$) |
> | --- | --- | --- |
> | BigSAN (w/ ReLU) | 6.20+/-0.27  | 9.16+/-0.08 |
> | BigSAN (w/ leakyReLU) | **5.97**+/-0.12 | **9.23**+/-0.03 |
>
> | CIFAR100 | FID ($\downarrow$) | IS ($\uparrow$) |
> | --- | --- | --- |
> | BigSAN (w/ ReLU) | 8.05+/-0.04  | 10.72+/-0.05 |
> | BigSAN (w/ leakyReLU) | **7.97**+/-0.06 | **10.84**+/-0.03 |

---

> > ### Comment · Reviewer_xjKk · 2023-11-20
> > **Reviewer reply**
> >
> > Thank you to the authors for their response. After reviewing it, I find that my concerns have been partially addressed. Therefore, I will maintain my initial score.

---

> > > ### Author Response · Authors · 2023-11-21
> > > **Author reply**
> > >
> > > Thank you for dedicating time to review our paper and for carefully checking our responses.
> > >
> > > In order to provide a comprehensive response to your questions and suggestions in your initial review, we would like to report an additional experiment on FFHQ-1024 using StyleGAN2 as below.
> > >
> > > We trained SAN by using the authors’ official repository. We obtained an FID score of 2.82, comparable to that of StyleGAN2 (2.84) as reported in the original paper [1]. However, we came across another study using the same repository, reporting a larger FID score (4.41) [2]. Currently, we are conducting an additional experiment to replicate the result of StyleGAN2. If we successfully reproduce the original score, we plan to include this comparison in the camera-ready manuscript.
> > >
> > > We hope we could have addressed all of your comments. If there are still unanswered questions or concerns, we would appreciate any additional feedback. Once again, thank you for taking the time to review our paper.
> > >
> > > [1] Karras et al., “Analyzing and Improving the Image Quality of StyleGAN,” CVPR 2020
> > >
> > > [2] Wang et al., “Diffusion-GAN: Training GANs with Diffusion,” ICLR, 2023

---

> > > > ### Author Response · Authors · 2023-11-24
> > > > **Author reply**
> > > >
> > > > It is fortunate that the discussion period for our paper has been extended! This allows us to report here that we have finished the reproduction experiment for StyleGAN2 on FFHQ (1024x1024), achieving a final FID score of 2.86. In summary, the FID score obtained by SAN (2.82) is comparable (slightly better) to that achieved by GAN.
> > > >
> > > > We appreciate your suggestion regarding the experiments on CIFAR-100 and FFHQ once again, as it allowed us to confirm the effectiveness of our proposal in broader scenarios. We observed that SAN consistently achieves better performance by a nontrivial margin in many cases. Furthermore, based on our observations, there is no case where converting GAN to SAN results in a performance deterioration. The only scenario where SAN is on par with GAN is observed in StyleGAN2 on FFHQ. These findings further support that SAN can serve as a drop-in replacement of GAN thanks to its universality.
> > > >
> > > > Now, we have completed all the suggested experiments. We hope all of your concerns have been properly addressed. We would appreciate it if you could provide further comments on our responses. Thanks for your devotion and constructive comments to our work.

---

> > > > > ### Author Response · Authors · 2023-11-28
> > > > > **Author reply**
> > > > >
> > > > > We express our sincere gratitude for your valuable comments. Currently, there are approximately 3 days left in the reviewer-author discussion period. We look forward to your response.

---

> > > > > ### Comment · Reviewer_xjKk · 2023-11-28
> > > > >
> > > > > Thanks to the authors for the thorough experimentation on FFHQ and CIFAR-100. All my concerns have been satisfactorily addressed, and I am pleased to support the acceptance of this paper.

---

> > > > > > ### Author Response · Authors · 2023-11-28
> > > > > > **Author reply**
> > > > > >
> > > > > > Thank you for your reply. We are glad to hear you support the acceptance.

---

### Author Response · Authors · 2023-11-17
**To all reviewers**

First of all, we would like to express our gratitude to all the reviewers for their time and effort in providing valuable feedback on our manuscript. We have carefully considered all comments and questions provided. Our responses are detailed below.

---
Besides, we would like to list not all but the major changes we have added to the manuscript. Please kindly refer to our response to each comment for our detailed changes. All the changes are colored in red in the revised manuscript.

**Figures**

- Added Figure 1 for illustrating the intuitions of *direction optimality* and *separability*

- Added Figure 3 for illustrating how each metrizable condition works in our theorem

**Additional experiments**

- Comparison of SAN and GAN on CIFAR-100 (Section 7.2)

- Fine-tuning of BigGAN with SAN (Appendix G)

- Empirically study on injectivity of activation function with BigGAN (Appendix H.3)

**Readability**

- Simplified some complicated notations

- Added Table 6 to provide readers a handy list of notations (Appendix A)

- Added sentences for providing intuitions of the metrizable conditions (around the first mentions).

---

### Meta-Review · Area_Chair_yFx1 · 2023-12-25

**Metareview:**

The paper addresses how inexact optimization in discriminator training influences gradient quality for generator training in GANs. The authors propose the concept of "metrizable discriminator", which refers to the discriminator for which the generator loss is a distance between two distributions. They showed that the metrizable condition can be satisfied if the three conditions hold: direction optimality, separability, and injectivity. They showed that Wasserstein GAN meets these conditions, whereas other GANs such as hinge-GAN do not. They propose a modification of GAN formulation termed SAN to make the formulation satisfy the direction optimality.
     The writing style presents some challenges to the reviewers. Key concepts, particularly the central "metrizable condition" is not well-explained beyond mathematical definitions, hindering comprehension. The reviewers have managed to grasp the main ideas, with considerable effort, and showed appreciation of the contributions. I have read the paper, and agree with the reviewers on the contributions. I also agree with the suggestions on improvement of writing (especially the missing connection to gradient-descent-ascent algorithm), and on adding explanations of the proof ideas in earlier works like Deshpande et al. It would be nice if the main ideas of the proofs (e.g. the main ideas of Theorem 3.5 and Proposition 4.5) can be explained in a simple section, possibly by explaining the main ideas of the proofs of Deshpande et al. or Kolouri et al.
      Overall, I recommend acceptance, but I strongly suggest the authors to include the above suggestions of the reviewers.

**Justification For Why Not Higher Score:**

Just resolved one subproblem of GANs.

**Justification For Why Not Lower Score:**

The problem of whether inexact discriminator can help generator training is an interesting problem.

---

### Decision · Program_Chairs · 2024-01-16

Accept (poster)